# Revealing Hidden Causal Variables and Latent Factors from Multiple Distributions

## Abstract

In many problems, the measured variables (e.g., image pixels) are just mathematical functions of the hidden causal variables (e.g., the underlying concepts or objects). For the purpose of making prediction in changing environments or making proper changes to the system, it is helpful to recover the hidden causal variables $Z_i$, their causal relations represented by graph $\mathcal{G}_Z$, and how their causal influences change, which can be explained by suitable latent factors $\theta_i$ governing changes in the causal mechanisms. This paper is concerned with the problem of estimating the underlying hidden causal variables and the latent factors from multiple distributions (arising from heterogeneous data or nonstationary time series) in nonparametric settings. We first show that under the sparsity constraint on the recovered graph over the latent variables and suitable sufficient change conditions on the causal influences, the recovered latent variables and their relations are related to the underlying causal model in a specific, nontrivial way. Moreover, we show that orthogonally, under the modular change condition on the causal modules (without the sparsity constraint on the graph), the underlying latent factors $\theta_i$ can be recovered up to component-wise invertible transformations. Putting them together, one is able to recover the hidden variables, their causal relations, and the corresponding latent factors up to minor indeterminacies.

## 1 Introduction

Causal representation learning holds paramount significance across numerous fields, offering insights into intricate relationships within datasets. Most traditional methodologies (e.g., causal discovery) assume the observation of causal variables. This assumption, however reasonable, falls short in complex scenarios involving indirect measurements, such as electronic signals, image pixels, and linguistic tokens. In addition, there are usually changes on the causal mechanisms in the real-world scenarios, such as the heterogeneous or nonstationary data. Identifying the hidden causal variables and their structures together with the change of the causal mechanism is in pressing need to understand the complicated real-world causal process.

At the same time, identifying only the hidden causal variables but not the structure among them, is already a considerable challenge. In the i.i.d. case, different latent representations can explain the same observations equally well, while not all of them are consistent with the true causal process. For instance, independent component analysis (ICA), where a set of observed variables $X$ is represented as a mixture of independent latent variables $Z$, i.e, $X = g(Z)$, is known to be unidentifiable without additional assumptions (Comon, 1994). While being a strictly easier task since there are no relations among hidden variables, the identifiability of ICA relies on conditions on distributional assumptions (non-i.i.d. data) (Hyvärinen & Morioka, 2016; 2017; Hyvärinen et al., 2019; Khemakhem et al., 2020a; Sorrenson et al., 2020; Lachapelle et al., 2022; Hälvä & Hyvärinen, 2020; Hälvä et al., 2021; Yao et al., 2022) or specific functional constraints (Comon, 1994; Hyvärinen & Pajunen, 1999; Taleb & Jutten, 1999; Buchholz et al., 2022; Zheng et al., 2022).

To generalize beyond the independent hidden variables and recover the causal structure among them, recent advances either introduce additional experiments in the forms of interventional or counterfactual data, or place more restrictive parametric or graphical assumptions on the latent causal model. For observational data, various graphical conditions have been proposed together with parametric assumptions such as linearity (Silva et al., 2006; Cai et al., 2019; Xie et al., 2020; 2022; Adams et al.,

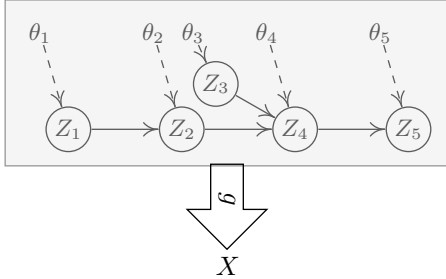

Figure 1: The generating process for each hidden causal variable $Z_i$ changes, governed by a latent factor $\theta_i$. The observations $X$ are generated by $X = g(Z)$ with a nonlinear mixing function $g$.

2021; Huang et al., 2022) and discreteness (Kivva et al., 2021). For interventional data, single-node interventions have been considered together with parametric assumptions (e.g., linearity) on the mixing funtion (Varici et al., 2023; Ahuja et al., 2023; Buchholz et al., 2022) or also on the latent causal model Squires et al. (2023). The nonparametric settings for both the mixing function and causal model have been explored by (Brehmer et al., 2022; von Kügelgen et al., 2023; Jiang & Aragam, 2023) together with additional assumptions on counterfactual views (Brehmer et al., 2022), distinct paired interventions (von Kügelgen et al., 2023), and graphical conditions (Jiang & Aragam, 2023).

However, while it is possible to discover the latent causal structure using additional assumptions, most previous studies fall short in identifying the potential shifts in the causal mechanism. In a constantly changing world, assuming a static hidden causal model can be counterintuitive. It is crucial to identify the latent factors that govern these changes. Consider, for instance, the essential characteristic of our brain, where neural connections are persistently evolving, leading to the typical nonstationarity of fMRI data (Havlicek et al., 2011). Regrettably, the majority of previous studies to reveal hidden causal variables and their structures do not consider recovering the latent factors directing the changes. As a result, current theories may not fully uncover the hidden causal model in complex real-world scenarios since the mechanisms of change remain elusive.

In this work, we aim to reveal the hidden causal variables, their causal relations, and the latent factors governing the changes in the causal mechanisms. Our results are only based on purely observational data, offering insight into the underlying causal models and changes without the need for experiments. We focus on the general nonparametric settings in the latent causal model, mixing function, and the influence of the change. Concretely, we show that under the sparsity constraint on the recovered graph and sufficient change on the causal influences, the ground-truth hidden variable can be identified in a specific way (Lemma 1, Thm. 2). Moreover, the causal relations among them are also recovered up to a certain indeterminacy with new relaxations of faithfulness (Lemma 1, Thm. 1, Thm. 3). Furthermore, we prove the component-wise identifiability for the latent factors governing the change, providing a nonparametric guarantees on the recovery of the hidden dynamics (Thm. 4). Therefore, we offer a collection of new identifiability findings in one of the most comprehensive settings, where not only the hidden causal variables and the structures but also the latent changing factors can be unveiled. Our theory has been validated by experiments across diverse settings.

## 2 PROBLEM SETTING

Let $X = (X_1, \ldots, X_d)$ be an $d$-dimensional random vector that represents the observations. We assume that they are generated by $n$ hidden causal variables $Z = (Z_1, \ldots, Z_n)$ via a nonlinear injective mixing function $g : \mathbb{R}^n \rightarrow \mathbb{R}^d$ $(d \geq n)$, which is also a $\mathcal{C}^2$ diffeomorphism. Furthermore, the variables $Z_i$'s are assumed to follow a structural equation model (SEM) (Pearl, 2000). Putting them together, the data generating process can be written as

$$\underbrace{X = g(Z)}_{\text{Nonlinear mixing}}, \quad \underbrace{Z_i = f_i(\text{PA}(Z_i), \epsilon_i; \theta_i), i = 1, \ldots, n}_{\text{Latent SEM}}. \tag{1}$$

where $\text{PA}(Z_i)$ denotes the parents of variable $Z_i$, $\epsilon_i$'s are exogenous noise variables that are mutually independent, and $\theta_i$ denotes the latent (changing) factor (or effective parameters) associated with each model. Here, the data generating process of each hidden variable $Z_i$ may change, e.g., across

domains or over time, governed by the corresponding latent factor $\theta_i$; it is commonplace to encounter such changes in causal mechanisms in practice (arising from heterogenous data or nonstationary time series). In addition, interventional data can be seen as a special type of change, which qualitatively restructure the causal relations. As their names suggest, we assume that the observations $X$ are observed, while the hidden causal variables $Z$ and latent factors $\theta = (\theta_1, \ldots, \theta_n)$ are unobserved.

Let $P_X$ and $P_Z$ be the distributions of $X$ and $Z$, respectively, and their corresponding probability density functions be $p_X(X; \theta)$ and $p_Z(Z; \theta)$, respectively. To lighten the notation, we drop the subscript in the density when the context is clear. The latent SEM in Eq. (1) induces a causal graph $\mathcal{G}_Z$ with vertices $\{Z_i\}_{i=1}^n$ and edges $Z_j \rightarrow Z_i$ if and only if $Z_j \in \text{PA}(Z_i)$. We assume that $\mathcal{G}_Z$ is acyclic, i.e., a directed acyclic graph (DAG). This implies that the distribution of variables $Z$ satisfy the Markov property w.r.t. DAG $\mathcal{G}_Z$ (Pearl, 2000), i.e., $p(Z; \theta) = \prod_{i=1}^n p(Z_i \mid \text{PA}(Z_i); \theta_i)$. We provide an example of the data generating process in Eq. (1) and its corresponding latent DAG $\mathcal{G}_Z$ in Figure 1. In particular, given the observations $X$ arising from multiple distributions (governed by the latent factors $\theta$), our goal is to recover the underlying hidden causal variables $Z = g^{-1}(X)$ and their causal relations, as well as the latent factors $\theta$, up to minor indeterminacies.

## 3 LEARNING CAUSAL REPRESENTATIONS WITH SPARSITY CONSTRAINTS

In this section, we provide theoretical results to show how one is able to recover the underlying hidden causal variables and their causal relations up to certain indeterminacies. Specifically, we show that under the sparsity constraint on the recovered graph over the latent variables and suitable sufficient change conditions on the causal influences, the recovered latent variables are related to the underlying hidden causal variables in a specific, nontrivial way. Such theoretical results serve as the foundation of our algorithm described in Section 5.

To start with, we estimate a model $(\hat{g}, \hat{f}, p_{\hat{Z}})$ which assumes the same data generating process as in Eq. (1) and matches the true distribution of $X$ in different domains:

$$p_X(X'; \theta') = p_{\hat{X}}(X'; \theta'), \quad \forall X', \theta'. \tag{2}$$

where $X$ and $\hat{X}$ are generated from the true model $(g, f, p_Z)$ and the estimated model $(\hat{g}, \hat{f}, p_{\hat{Z}})$, respectively.

A key ingredient in our theoretical analysis is Markov network that represent conditional dependencies among random variables in a graphical manner via an undirected graph. Let $\mathcal{M}_Z$ be the Markov network over variables $Z$, specifically, with vertices $\{Z_i\}_{i=1}^n$ and edges $(i, j) \in \mathcal{E}(\mathcal{M}_Z)$ if and only if $Z_i \not\perp\!\!\!\perp Z_j \mid Z_{[n] \setminus \{i, j\}}$.[1] Also, we denote by $|\mathcal{M}_Z|$ the number of undirected edges in the Markov network. In Section 3.1, apart from showing how to estimate the underlying hidden causal variables up to certain indeterminacies, we also show that such latent Markov network $\mathcal{M}_Z$ can be recovered up to trivial indeterminacies (i.e., relabeling of the hidden variables). To achieve so, we make use of the following property (assuming that $p_Z$ is twice differentiable):

$$Z_i \perp\!\!\!\perp Z_j \mid Z_{[n] \setminus \{i, j\}} \iff \frac{\partial^2 \log p(Z)}{\partial Z_i \partial Z_j} = 0.$$

Such a connection between pairwise conditional independence and cross derivatives of the density function has been noted by Lin (1997). With the recovered latent Markov network structure, we provide results in Section 3.2 to show how it relates to the true latent causal DAG $\mathcal{G}_Z$, by exploiting a specific type of faithfulness assumption that is considerably weaker than the standard faithfulness assumption used in the literature of causal discovery (Spirtes et al., 2001).

### 3.1 RECOVERING HIDDEN CAUSAL VARIABLES AND LATENT MARKOV NETWORK

We show how one benefits from multiple distributions to recover the hidden causal variables and the true Markov network structure among them up to minor indeterminacies, by making use of sparsity constraint and sufficient change conditions on the causal mechanisms.

We start with the following result that provides information about the relationship between the Markov network $\mathcal{M}_Z$ over true hidden causal variables $Z$ and the Markov network $\mathcal{M}_{\hat{Z}}$ over the estimated hidden variables $\hat{Z}$. This result serves as the backbone of our further analysis in this section. Denote by $\oplus$ the vector concatenation symbol.

---

[1] We use $[n]$ to denote $\{1, \ldots, n\}$ and $Z_{[n] \setminus \{i, j\}}$ to denote $\{Z_i\}_{i=1}^n \setminus \{Z_i, Z_j\}$.

**Lemma 1.** *Let the observations be sampled from the data generating process in Eq. (1), and $\mathcal{M}_Z$ be the Markov network over $Z$. Suppose that the following assumptions hold:*

- *A1 (Smooth and positive density): The probability density function of latent causal variables is smooth and positive, i.e. $p_Z$ is smooth and $p_Z > 0$ over $\mathbb{R}^n$.*

- *A2 (Sufficient changes): For any $Z \in \mathbb{R}^n$, there exist $2n + |\mathcal{M}_Z| + 1$ values of $\theta$, i.e., $\theta^{(u)}$ with $u = 0, \ldots, 2n + |\mathcal{M}_Z|$, such that the vectors $w(Z, u) - w(z, 0)$ with $u = 1, \ldots, 2n + |\mathcal{M}_Z|$ are linearly independent, where vector $w(Z, u)$ is defined as*

$$w(Z, u) = \left( \frac{\partial \log p(Z; \theta^{(u)})}{\partial Z_1}, \ldots, \frac{\partial \log p(Z; \theta^{(u)})}{\partial Z_n}, \right.$$
$$\left. \frac{\partial^2 \log p(Z; \theta^{(u)})}{\partial Z_1^2}, \ldots, \frac{\partial^2 \log p(Z; \theta^{(u)})}{\partial Z_n^2} \right)$$
$$\oplus \left( \frac{\partial^2 \log p(Z; \theta^{(u)})}{\partial Z_i \partial Z_j} \right)_{(i,j) \in \mathcal{E}(\mathcal{M}_Z)}.$$

*Suppose that we learn $(\hat{g}, \hat{f}, p_{\hat{Z}})$ to achieve Eq. (2). Then, for every pair of estimated hidden variables $\hat{Z}_k$ and $\hat{Z}_l$ that are not adjacent in the Markov network $\mathcal{M}_{\hat{Z}}$ over $\hat{Z}$, we have the following statements:*

- *(a) Each true hidden causal variable $Z_i$ is a function of at most one of $\hat{Z}_k$ and $\hat{Z}_l$.*
- *(b) For each pair of true hidden causal variables $Z_i$ and $Z_j$ that are adjacent in the Markov network $\mathcal{M}_Z$ over $Z$, at most one of them is a function of $\hat{Z}_k$ or $\hat{Z}_l$.*

The proof is provided in Appx. A. It is worth noting that the requirement of a sufficient number of environments has been commonly adopted in the literature (e.g., see (Hyvärinen et al., 2023) for a recent survey), such as visual disentanglement (Khemakhem et al., 2020b), domain adaptation (Kong et al., 2022), video analysis (Yao et al., 2021), and image-to-image translation (Xie et al., 2023). Also, we do not specify exactly how to learn $(\hat{g}, \hat{f}, p_{\hat{Z}})$ to achieve Eq. (2), and leave the door open for different approaches to be used, such as normalizing flow or variational approaches. For example, we adopt a variational approach in Section 5.

The above result sheds light on how each pair of estimated latent variables $\hat{Z}_k$ and $\hat{Z}_l$ that are not adjacent in Markov network $\mathcal{M}_{\hat{Z}}$ relate to the true hidden causal variables $Z$. Without any constraint on the estimating process, a trivial solution would be a complete graph over $\hat{Z}$. To avoid it, we enforce sparsity of the Markov network over $\hat{Z}$. This allows us to recover the true Markov network structure up to trivial indeterminacies, formally stated below, with a proof provided in Appx. B.

**Theorem 1** (Identifiability of Latent Markov Network). *Let the observations be sampled from the data generating process in Eq. (1), and $\mathcal{M}_Z$ be the Markov network over $Z$. Suppose that Assumptions A1 and A2 from Theorem 1 holds. Suppose also that we learn $(\hat{g}, \hat{f}, p_{\hat{Z}})$ to achieve Eq. (2) with the minimal number of edges of Markov network $\mathcal{M}_{\hat{Z}}$ over $\hat{Z}$. Then, the Markov network $\mathcal{M}_{\hat{Z}}$ over estimated hidden variables $\hat{Z}$ is isomorphic to the true latent Markov network $\mathcal{M}_Z$.*

Apart from recovering the true Markov network $\mathcal{M}_Z$, we show that the sparsity constraint on the Markov network structure over $\hat{Z}$ also allows us to recover the underlying hidden causal variables $Z$ up to minor indeterminacies.

**Theorem 2** (Identifiability of Hidden Causal Variables). *Let the observations be sampled from the data generating process in Eq. (1), and $\mathcal{M}_Z$ be the Markov network over $Z$. Let $N_{Z_i}$ be the set of neighbors of variable $Z_i$ in $\mathcal{M}_Z$. Suppose that Assumptions A1 and A2 from Theorem 1 holds. Suppose also that we learn $(\hat{g}, \hat{f}, p_{\hat{Z}})$ to achieve Eq. (2) with the minimal number of edges of Markov network $\mathcal{M}_{\hat{Z}}$ over $\hat{Z}$. Then, there exists a permutation $\pi$ of the estimated hidden variables, denoted as $\hat{Z}_\pi$, such that the corresponding statements hold for $i = 1, \ldots, n$:*

- *(a) $Z_i$ is a function of $\hat{Z}_{\pi(i)}$ and a (possibly empty) subset of the variables in $\{\hat{Z}_{\pi(j)} \mid Z_j \text{ is adjacent to } Z_i \text{ and all other neighbors of } Z_i \text{ in } \mathcal{M}_Z\}$.*

(b) $\hat{Z}_{\pi(i)}$ is a function of $Z_i$ and a (possibly empty) subset of the variables in $\{Z_j \mid Z_j \text{ is adjacent to } Z_i \text{ and all other neighbors of } Z_i \text{ in } \mathcal{M}_Z\}$.

The proof is given in Appx. C. It is worth noting that in many cases, the above result already enables us to recover some of the hidden variables up to a component-wise transformation.

**Remark 1.** *No matter how many neighbors each hidden causal variable $Z_i$ has, as long as one of its neighbors is not adjacent to any other neighbor in the Markov network $\mathcal{M}_Z$, $Z_i$ can be recovered up to a component-wise transformation.*

Even if the above case does not hold, Theorem 2 still shows how the estimated hidden variables relate to the underlying causal variables in a specific, nontrivial way. An example is provided below.

**Example 1.** *Consider the Markov network $\mathcal{M}_Z$ corresponding to the DAG $\mathcal{G}_Z$ over $Z_i$ in Figure 1. By Theorem 2 and suitable permutation of estimated hidden variables $\hat{Z}$, we have: (a) $\hat{Z}_{\pi(1)}$ is a function of $Z_1$ and possibly $Z_2$, (b) $\hat{Z}_{\pi(2)}$ is a function of $Z_2$, (c) $\hat{Z}_{\pi(3)}$ is a function of $Z_3$ and possibly $Z_2, Z_4$, (d) $\hat{Z}_{\pi(4)}$ is a function of $Z_4$, and (d) $\hat{Z}_{\pi(5)}$ is a function of $Z_5$ and possibly $Z_4$.*

In the example above, the hidden causal variables $Z_2$ and $Z_4$ can be recovered up to component-wise transformation, while variables $Z_1$, $Z_3$, and $Z_5$ can be identified up to mixtures with certain neighbors in the Markov network.

**Permutation of estimated latent variables.** Theorems 1 and 2 involve certain permutation of the estimated hidden variables $\hat{Z}$. Such an indeterminacy is common in the literature of causal discovery and representation learning tasks involving latent variables. In our case, since the function $h := \hat{g}^{-1} \circ g$ where $\hat{Z} = h(Z)$ is invertible, there exists a permutation of the latent variables such that the corresponding Jacobian matrix $J_h$ has nonzero diagonal entries (see Lemma 3 in Appx. B); such a permutation is what Theorems 1 and 2 refer to.

## 3.2 FROM LATENT MARKOV NETWORK TO LATENT CAUSAL DAG

Now we have identified the Markov network up to an isomorphism, which characterizes conditional independence relations in the distribution. To build the connection between Markov network or conditional independence relations and causal structures, prior theory replies on the Markov and faithfulness assumptions. However, in real-world scenarios, the faithfulness assumption could be violated due to various reasons including path cancellations (Zhang & Spirtes, 2008; Uhler et al., 2013).

Since our goal is to generalize the identifiability theory as much as possible to fit practical applications, we introduce two relaxations of the faithfulness assumptions:

**Assumption 1** (Single adjacency-faithfulness (SAF)). *Given a DAG $\mathcal{G}_Z$ and distribution $P_Z$ over the variable set $Z$, if two variables $Z_i$ and $Z_j$ are adjacent in $\mathcal{G}_Z$, then $Z_i \not\perp\!\!\!\perp Z_j | Z_{[n] \setminus \{i,k\}}$.*

**Assumption 2** (Single unshielded-collider-faithfulness (SUCF) (Ng et al., 2021)). *Given a latent causal graph $\mathcal{G}_Z$ and distribution $P_Z$ over the variable set $Z$, let $Z_i \rightarrow Z_j \leftarrow Z_k$ be any unshielded collider in $\mathcal{G}_Z$, then $Z_i \not\perp\!\!\!\perp Z_k | Z_{[n] \setminus \{i,k\}}$.*

We propose SAF as a relaxtion of the Adjacency-faithfulness (Ramsey et al., 2012). The SUCF assumption is first introduced by Ng et al. (2021), which is strictly weaker than Orientation-faithfulness (Ramsey et al., 2012). Thus, both of them are strictly weaker than the faithfulness assumption, since the combination of Adjacency-faithfulness and Orientation-faithfulness is weaker than the faithfulness assumption (Zhang & Spirtes, 2008).

Interestingly, not only they are weaker variants of faithfulness, we prove that they are actually necessary and sufficient conditions, thus the weakest possible ones, to bridge conditional independence relations and causal structures. Specifically, we show that the recovered Markov network is exactly the moralized graph of the true causal DAG if and only if the proposed variants of faithfulness hold. The proofs of Lemma 2 and Theorem 3 are shown in Appx. D.

**Lemma 2.** *Given a latent causal graph $\mathcal{G}_Z$ and distribution $P_Z$ with its Markov Network $\mathcal{M}_Z$, under Markov assumption, the undirected graph defined by $\mathcal{M}_Z$ is a subgraph of the moralized graph of the true causal DAG $G$.*

**Theorem 3.** *Given a causal DAG $\mathcal{G}_Z$ and distribution $P_Z$ with its Markov Network $\mathcal{M}_Z$, under Markov assumption, the undirected graph defined by $\mathcal{M}_Z$ is the moralized graph of the true causal DAG $\mathcal{G}_Z$ if and only if the SAF and SUCF assumptions are satisfied.*

It is worth noting that the connection between conditional independence relations and causal structures has been developed by (Loh & Bühlmann, 2014; Ng et al., 2021) in the linear case by leveraging the properties of the inverse covariance matrix; our results here focus on the nonparametric case and thus being able to serve the considered general settings for identifiability. The necessary and sufficient assumptions may also be of independent interest for other causal discovery tasks exploring conditional independence relations.

## 4 LEARNING LATENT FACTORS GOVERNING THE CHANGES

In previous section, we describe how the sparsity constraint on recovered graph over latent variables and sufficient change conditions on causal influences can be used to recover the latent variables and causal graph up to certain indeterminacies. In addition to hidden variables and causal graph, one may wonder whether it is possible to recover the latent factors that govern the changes across multiple distributions (e.g., arising from heterogeneous data or nonstationary time series). The changing parameters of different distributions of interest often lie in low-dimensional manifold, which leads to a low-dimensional and interpretable representation of changing mechanisms (Stojanov et al., 2019). We show that under the modular changes condition on the causal modules (without sparsity constraint on the graph), the latent factors $\theta_i$ can be recovered up to component-wise invertible transformations.

**Theorem 4** (Identifiability of Latent Factors). *Let the observations be sampled from the data generating process in Eq. (1), and $\mathcal{M}_Z$ be the Markov network over $Z$. Suppose that the following assumptions hold:*

- *A1 (Smooth and positive density): The probability density function of latent causal variables is smooth and positive, i.e. $p_Z$ is smooth and $p_Z > 0$ over $\mathbb{R}^n$.*

- *A2 (Modular changes): $\theta_i$ across different $i$ are not related via equality constraints.*

- *A3 (Sufficient changes): for $Z \in \mathbb{R}^n$ and each value of $\theta$, the vectors $w(\theta)$ are linearly independent, where $w(\theta)$ is defined as*

$$w(\theta) = \left( \frac{\partial \log p(Z_1 \mid PA(Z_1); \theta_1)}{\partial \theta_1}, \ldots, \frac{\partial \log p(Z_n \mid PA(Z_n); \theta_n)}{\partial \theta_n}, \right.$$
$$\left. \frac{\partial^2 \log p(Z_1 \mid PA(Z_1); \theta_1)}{\partial \theta_1^2}, \ldots, \frac{\partial^2 \log p(Z_n \mid PA(Z_n); \theta_n)}{\partial \theta_n^2} \right).$$

*Suppose that we learn $(\hat{g}, \hat{f}, p_{\hat{Z}})$ to achieve Eq. (2). Then, $\theta_i$ are identifiable up to component-wise invertible transformation from the observations $X$ in multiple domains.*

The proof is provided in Appx. E. The condition of modular (or independent) changes above avoids the latent factors $\theta$ to be coupled in a specific way. It is worth noting that similar principles of modular changes have been adopted in the literature of causal discovery (Huang et al., 2020) to learn causal structure from multiple distributions without hidden variables.

### 4.1 BENEFIT FROM PARAMETRIC CONSTRAINTS ON CHANGES

In several cases, we may be able to leverage the parametric constraint on the changes for the recovery of the model. For instance, if we know that the changes happen to the linear causal mechanisms with Gaussian noises, this constraint can immediately help reduce the search space and improve the identifiability. As an illustrative example, consider a linear Gaussian causal model for the latent variables with structure $Z_1 \rightarrow Z_2$. In the true model, only limited parameters change according to the latent factor, while for the alternative model, the change of parameters follow equality constraints and thus all parameters can change. Thus, the identifiability is provided by assuming parametric constraints such as the linear Gaussian model, which may be helpful in certain applications.

## 5 CHANGE ENCODING NETWORK FOR REPRESENTATION LEARNING

Thanks to the identifiability result, we now present two different practical implementations to recover the latent variables and their causal relations from observations from multiple domains. We build our method on the variational autoencder (VAE) framework and can be easily extended to other models, such as normalizing flows.

We learn a deep latent generative model (decoder) $p(X|Z)$ and a variational approximation (encoder) $q(Z|X, \theta)$ of its true posterior $p(Z|X, \theta)$ since the true posterior is usually intractable. To learn the model, we minimize the lower bound of the log-likelihood as

$$\log p(X|u, \theta) = \log \int p(X|Z, u)p(Z|u)dZ \tag{3}$$
$$= \log \int \frac{q(Z|X, u)}{q(Z|X, u)} \int p(X|Z, u, \theta)p(Z|u)dZ$$
$$\geq -\text{KL}(q(Z|X, u)||p(Z|u)) + \mathbb{E}_q[\log p(X|Z)].$$

For the posterior $q(Z|X, u)$, we assume that it is a multivariate Gaussian or a Laplacian distribution, where the mean and variance are generated by the neural network encoder. As for $q(X|Z)$, we assume that it is a multivariate Gaussian and the mean is the output of the decoder and the variance is a pre-defined value (we use 0.01). Besides this lower bound likelihood, we use an adjacency matrix $\hat{A}$ to capture the relationship among the estimated latent variables. According to our theoretical results, we apply a sparsity constraint to avoid trivial solution and find the true structure. Therefore, we apply $\ell_1$ regularization to learn a graph which is sparse but sufficient to explain the changes across components and domains.

An essential difference from VAE (Kingma & Welling, 2013) and iVAE (Khemakhem et al., 2020a) is that our method allow the components of $Z$ to be causally dependent and we are able to learn the components and causal relationships. And the key is the prior distribution $P(Z|u)$. Now we present two different implementations to capture the changes with a properly defined prior distribution.

### 5.1 NON-PARAMETRIC IMPLEMENTATION

To recover the relationships and latent variables $Z$, we build the normalizing flow to mimic the inverse of the latent SEM $Z_i = f_i(\text{PA}(Z_i), \epsilon_i)$ in Eq. (1). We first assume a causal ordering as $\hat{Z}_1, \ldots, \hat{Z}_n$. Then, for each component $\hat{Z}_i$, we consider the previous components $\{\hat{Z}_1, \ldots, \hat{Z}_{i-1}\}$ as potential parents of $\hat{Z}_i$ and we can select the true parents with the adjacency matrix $\hat{A}$, where $\hat{A}_{i,j}$ denotes that component $\hat{Z}_j$ contributes in the generation of $\hat{Z}_i$. If $\hat{A}_{i,j} = 0$, it means that $\hat{Z}_j$ will not contribute to the generation of $\hat{Z}_i$. Then, we use the selected parents $\{\hat{A}_{i,1}\hat{Z}_1, \ldots, \hat{A}_{i,i-1}\hat{Z}_{i-1}\}$ and the domain label $u$ to generate parameters of normalizing flow and apply the flow transformation on $\hat{Z}_i$ to turn it into $\hat{\epsilon}_i$. Specifically, we have

$$\hat{\epsilon}_i, \log \det_i = \text{Flow}(\hat{Z}_i; \text{NN}(\{\hat{A}_{i,j}\hat{Z}_j\}_{j=1}^{i-1}, u)), \tag{4}$$

where $\log \det_i$ is the log determinant of the flow transformation on $\hat{Z}_i$.

To compute the prior distribution, we make assumption on the noise term $\epsilon$ that it follows an independent prior distribution $p(\epsilon)$, such as a standard isotropic Gaussian or a Laplacian. Then according to the change of variable formula, the prior distribution of the dependent latents can be written as

$$\log p(\hat{Z}|u) = \log p(\hat{\epsilon}) + \sum_{i=1}^{n} \log \det_i. \tag{5}$$

Intuitively, to minimize the KL divergence loss between $p(Z|u)$ and $q(Z|X, u)$, the network has to learn the correct structure and the underlying latent variables; otherwise, it can be difficult to transform the dependent latent variables $\hat{Z}$ to a factorized prior distribution, e.g., $\mathcal{N}(0, \text{I})$.

### 5.2 PARAMETRIC IMPLEMENTATION

As mentioned in Section 4.1, we can make parametric assumption on the latent causal process and facilitate the learning of true causal structure and components. Here, we consider the linear SEM

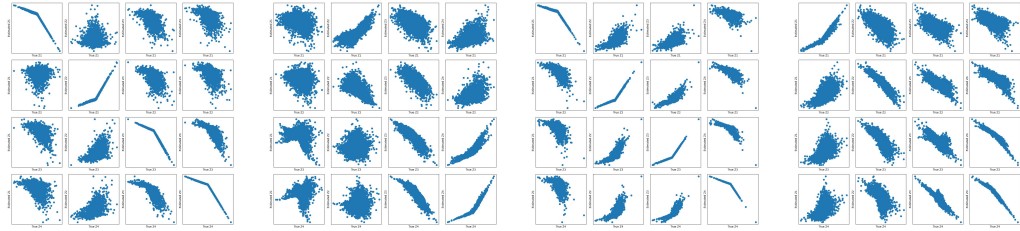

Figure 2: Recovered latent variables v.s. the true latent variables with Non-Parametric Approach. (a) Y-structure with Laplace noise. (b) Y-structure with Gaussian noise. (c) Chain structure with Laplace noise. (d) Chain structure with Gaussian noise.

and more complex SEMs can be generalized. Specifically, we assume

$$Z = A(C^{(u)}Z) + S^{(u)}\epsilon + B^{(u)}, \tag{6}$$

where $A$ is a causal adjacency matrix which can be permuted to be strictly lower-triangular, $C^{(u)}$ and $S^{(u)}$ are underlying domain-specific scaling matrix and vector, respectively, $B^{(u)}$ is the underlying domain-specific bias vector, and $\epsilon$ is the independent noise. $\theta$ governs the scaling and bias values.

To estimate the latent variables $Z$, the causal structure $A$, and capture the changes across domains, we introduce the learnable scaling and bias parameters and pre-define a causal ordering as $\hat{Z}_1, \hat{Z}_2, \ldots, \hat{Z}_n$. Then we have the matrix form as

$$\hat{\epsilon} = (\hat{Z} - \hat{B}^{(u)} - \hat{A}\hat{C}^{(u)}\hat{Z})/\hat{S}^{(u)}. \tag{7}$$

Given a prior distribution $p(\hat{\epsilon})$, and according to the change of variable rule, we have the prior distribution for $\hat{Z}$ as

$$\log p(\hat{Z}|u) = \log p(\hat{\epsilon}) - \log \left| \sum_{i=1}^{n} \hat{S}_i^{(u)} \right|. \tag{8}$$

We then minimize the lower bound in Eq. (4).

### 5.3 SIMULATIONS

To verify our theory and the proposed implementations, we run experiments on the simulated data because the ground truth causal adjacency matrix and the latent variables across domains are available for simulated data. Consequently, we consider following common causal structures (i) Y-structure with 4 variables, $Z_1 \to Z_3 \leftarrow Z_2, Z_3 \to Z_4$ and (ii) chain structure $Z_1 \to Z_2 \to Z_3 \to Z_4$. The noises are modulated with scaling random sampled from $\text{Unif}[0.5, 2]$ and biases are sampled from $\text{Unif}[-2, 2]$. The scaling on the $Z$ are also randomly sampled from $\text{Unif}[0.5, 2]$. In other words, the changes are modular. After generating $Z$, we feed the latent variables into MLP with orthogonal weights and LeakyReLU activations for invertibility. We present the results in Fig. 2 and 3. Firstly, we observe that all latent variables can be recovered accurately in Y and chain structures with the linear parameterization implementation. The hidden structure is also recovered. This supports our theoretical result that the components and structure are identifiable up to certain indeterminacies. As for the results in Fig. 2, we observe that our non-parametric method is still able to recover the true latent variables with Laplace noise. However, when the noises are Gaussian, it becomes more challenging and we observe that some components are mixed, which further aligns with our theory (e.g., Theorem 2) and demonstrate the benefit of using suitable parameterization.

## 6 RELATED WORK

Causal representation learning aims to unearth causal latent variables and their relations from observed data. Despite its significance, the identifiability of the hidden generating process is known to be impossible without additional constraints, especially with only observational data. In the linear, non-Gaussian case, Silva et al. (2006) recover the Markov equivalence class, provided that each observed variable has a unique latent causal parent; Xie et al. (2020); Cai et al. (2019) estimate the latent variables and their relations assuming at least twice measured variables as latent ones, which has been further extended to learn the latent hierarchical structure (Xie et al., 2022). Moreover,

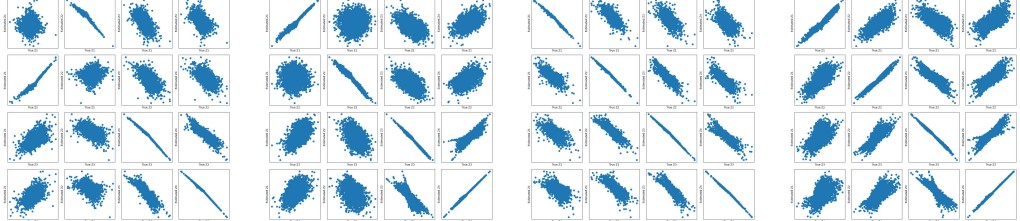

Figure 3: Recovered latent variables v.s. the true latent variables with Linear Parameterization Approach. The $X$-axis denotes the components of true latent variables $Z$ and the $Y$-axis represent the components of estimated latent variables $\hat{Z}$. (a) Y-structure with Laplace noise. (b) Y-structure with Gaussian noise. (c) Chain structure with Laplace noise. (d) Chain structure with Gaussian noise.

Adams et al. (2021) provide theoretical results on the graphical conditions for identification. In the linear, Gaussian case, Huang et al. (2022) leverage rank deficiency of the observed sub-covariance matrix to estimate the latent hierarchical structure. In the discrete case, Kivva et al. (2021) identify the hidden causal graph up to Markov equivalence by assuming a mixture model where the observed children sets of any pair of latent variables are different.

Given the challenge of identifiability on purely observational data, a different line of research leverage experiments by assuming the accessibility of various types of interventional data. Based on the single-node perfect intervention, Squires et al. (2023) leverage single-node interventions for the identifiability of linear causal model and linear mixing function; (Varici et al., 2023) incorporate for nonlinear causal model and linear mixing function; (Varici et al., 2023; Buchholz et al., 2023; Jiang & Aragam, 2023) provide the identifiability of the nonparametric causal model and linear mixing function; (Ahuja et al., 2023) further generalize the result to nonparametric causal model and polynomial mixing functions with additional constraints on the latent support; and (Brehmer et al., 2022; von Kügelgen et al., 2023; Jiang & Aragam, 2023) explore the nonparametric settings for both the causal model and mixing function. In addition to the single-node perfect interventions, Brehmer et al. (2022) introduced counterfactual pre- and post-intervention views; von Kügelgen et al. (2023) assume two distinct, paired interventions per node for multivariate causal models; and Jiang & Aragam (2023) places specific structural restrictions on the latent causal graph.

Our study lies in the line of leveraging only observational data, and provides a set of identifiability results in the general nonparametric settings on *both* the latent causal model and mixing function. Unlike prior works with observational data, we do not have any parametric assumptions or graphical restrictions; Compared to those relying on interventional data, our results naturally benefit from the heterogeneity of observational data (e.g., multi-domain data, nonstationary time series) and avoid additional experiments for interventions. In addition, we recover the latent factors governing the change up to trivial indeterminacies, shedding new light on understanding the dynamics in the latent causal process, a view point overlooked by most existing work.

## 7 CONCLUSION AND DISCUSSIONS

We establish a set of new identifiability results to reveal hidden causal variables, latent structures, and latent factors governing the changes of the causal mechanisms in the general nonparametric settings. Specifically, with sparsity regularization during estimation and sufficient changes in the causal influences, we demonstrate that the revealed hidden variables and structures are related to the underlying causal model in a specific, nontrivial way. Furthermore, we prove that the latent factors governing the change in the causal mechanism can be identified up to trivial indeterminacies. In contrast to recent works on the recovery of hidden causal variables and structures, our results rely on purely observational data without graphical or parametric constraints. Additionally, the identification of latent changing factors illustrate the potential to fully comprehend the hidden world, as even the latent dynamics could also be unearthed. Therefore, our results offer insight into unveiling the latent causal process in one of the most universal settings. Experiments in various settings have been conducted to validate the theory. As future work, we will explore the scenario where only a subset of the causal relations change, which could be a challenge as well as a chance, and show up to what extent the underlying causal variables can be recovered with potentially weaker assumptions.

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

# A    PROOF OF LEMMA 1

**Lemma 1.** *Let the observations be sampled from the data generating process in Eq. (1), and $\mathcal{M}_Z$ be the Markov network over $Z$. Suppose that the following assumptions hold:*

- *A1 (Smooth and positive density): The probability density function of latent causal variables is smooth and positive, i.e. $p_Z$ is smooth and $p_Z > 0$ over $\mathbb{R}^n$.*

- *A2 (Sufficient changes): For any $Z \in \mathbb{R}^n$, there exist $2n + |\mathcal{M}_Z| + 1$ values of $\theta$, i.e., $\theta^{(u)}$ with $u = 0, \ldots, 2n + |\mathcal{M}_Z|$, such that the vectors $w(Z, u) - w(z, 0)$ with $u = 1, \ldots, 2n + |\mathcal{M}_Z|$ are linearly independent, where vector $w(Z, u)$ is defined as*

$$w(Z, u) = \left( \frac{\partial \log p(Z; \theta^{(u)})}{\partial Z_1}, \ldots, \frac{\partial \log p(Z; \theta^{(u)})}{\partial Z_n}, \right.$$
$$\left. \frac{\partial^2 \log p(Z; \theta^{(u)})}{\partial Z_1^2}, \ldots, \frac{\partial^2 \log p(Z; \theta^{(u)})}{\partial Z_n^2} \right)$$
$$\oplus \left( \frac{\partial^2 \log p(Z; \theta^{(u)})}{\partial Z_i \partial Z_j} \right)_{(i,j) \in \mathcal{E}(\mathcal{M}_Z)}.$$

*Suppose that we learn $(\hat{g}, \hat{f}, p_{\hat{Z}})$ to achieve Eq. (2). Then, for every pair of estimated hidden variables $\hat{Z}_k$ and $\hat{Z}_l$ that are not adjacent in the Markov network $\mathcal{M}_{\hat{Z}}$ over $\hat{Z}$, we have the following statements:*

- *(a) Each true hidden causal variable $Z_i$ is a function of at most one of $\hat{Z}_k$ and $\hat{Z}_l$.*
- *(b) For each pair of true hidden causal variables $Z_i$ and $Z_j$ that are adjacent in the Markov network $\mathcal{M}_Z$ over $Z$, at most one of them is a function of $\hat{Z}_k$ or $\hat{Z}_l$.*

*Proof.* Since $h$ in $\hat{Z} = h(Z)$ is invertible, we have

$$p(\hat{Z}; \hat{\theta}) = p(Z; \theta)/|J_h|,$$
$$\log p(\hat{Z}; \hat{\theta}) = \log p(Z; \theta) - \log |J_h|. \tag{9}$$

Suppose $\hat{Z}_k$ and $\hat{Z}_l$ are conditionally independent given $\hat{Z}_{[n] \setminus \{k,l\}}$ i.e., they are not adjacent in the Markov network over $\hat{Z}$. For each $\theta$, by Lin (1997), we have

$$\frac{\partial^2 \log p(\hat{Z}; \hat{\theta})}{\partial \hat{Z}_k \partial \hat{Z}_l} = 0.$$

To see what it implies, we find the first-order derivative:

$$\frac{\partial \log p(\hat{Z}; \theta)}{\partial \hat{Z}_k} = \sum_i \frac{\partial \log p(Z; \theta)}{\partial Z_i} \frac{\partial Z_i}{\partial \hat{Z}_k} - \frac{\partial \log |J_q|}{\partial \hat{Z}_k}.$$

Let $\eta(\theta) = \log p(Z; \theta)$, $\eta_i'(\theta) = \frac{\partial \log p(Z;\theta)}{\partial Z_i}$, $\eta_{ij}''(\theta) = \frac{\partial^2 \log p(Z;\theta)}{\partial Z_i \partial Z_j}$, $h_{i,l}' = \frac{\partial Z_i}{\partial \hat{Z}_l}$, and $h_{i,kl}'' = \frac{\partial^2 Z_i}{\partial \hat{Z}_k \partial \hat{Z}_l}$. We then derive the second-order derivative:

$$0 = \sum_j \sum_i \frac{\partial^2 \log p(Z; \theta)}{\partial Z_i \partial Z_j} \frac{\partial Z_j}{\partial \hat{Z}_l} \frac{\partial Z_i}{\partial \hat{Z}_k} + \sum_i \frac{\partial \log p(Z; \theta)}{\partial Z_i} \frac{\partial^2 Z_i}{\partial \hat{Z}_k \partial \hat{Z}_l} - \frac{\partial^2 \log |J_q|}{\partial \hat{Z}_k \partial \hat{Z}_l}.$$

$$= \sum_i \frac{\partial^2 \log p(Z; \theta)}{\partial Z_i^2} \frac{\partial Z_i}{\partial \hat{Z}_l} \frac{\partial Z_i}{\partial \hat{Z}_k} + \sum_j \sum_{(j,i) \in \mathcal{E}(\mathcal{M}_Z)} \frac{\partial^2 \log p(Z; \theta)}{\partial Z_i \partial Z_j} \frac{\partial Z_j}{\partial \hat{Z}_l} \frac{\partial Z_i}{\partial \hat{Z}_k}$$

$$+ \sum_i \frac{\partial \log p(Z; \theta)}{\partial Z_i} \frac{\partial^2 Z_i}{\partial \hat{Z}_k \partial \hat{Z}_l} - \frac{\partial^2 \log |J_q|}{\partial \hat{Z}_k \partial \hat{Z}_l}$$

$$= \sum_i \eta_{ii}''(\theta) h_{i,l}' h_{i,k}' + \sum_j \sum_{(j,i) \in \mathcal{E}(\mathcal{M}_Z)} \eta_{ij}''(\theta) h_{j,l}' h_{i,k}' + \sum_i \eta_i'(\theta) h_{i,kl}'' - \frac{\partial^2 \log |J_q|}{\partial \hat{Z}_k \partial \hat{Z}_l}. \tag{10}$$

Here we denote by $\mathcal{E}(\mathcal{M}_Z)$ the set of edges in the Markov network over $Z$ and we already make use of the fact that if $Z_i$ and $Z_j$ are not adjacent in the Markov network, then $\frac{\partial^2 \log p(Z;\theta)}{\partial Z_i \partial Z_j} = 0$.

By Assumption A2, consider the $2n + |\mathcal{M}_Z| + 1$ values of $\theta$, i.e., $\theta^{(u)}$ with $u = 0, \ldots, 2n + |\mathcal{M}_Z|$, such that Eq. (10) hold. Then, we have $2n + |\mathcal{M}_Z| + 1$ such equations. Subtracting each equation corresponding to $\theta^{(u)}, u = 1, \ldots, 2n + |\mathcal{M}_Z|$ with the equation corresponding to $\theta^{(0)}$ resuls in $2n + |\mathcal{M}_Z|$ equations:

$$0 = \sum_i (\eta_{ii}''(\theta^{(u)}) - \eta_{ii}''(\theta^{(0)}))h_{i,l}'h_{i,k}' + \sum_j \sum_{(j,i) \in \mathcal{E}(\mathcal{M}_Z)} (\eta_{ij}''(\theta^{(u)}) - \eta_{ij}''(\theta^{(0)}))h_{j,l}'h_{i,k}'$$
$$+ \sum_i (\eta_i'(\theta^{(u)}) - \eta_i'(\theta^{(0)}))h_{i,kl}'',$$

where $u = 1, \ldots, 2n + |\mathcal{M}_Z|$. By Assumption A2, the vectors formed by collecting the corresponding coefficients are linearly independent. Therefore, for any $i$ and any $j$ such that $(j, i) \in \mathcal{E}(\mathcal{M}_Z)$, we have

$$h_{i,l}'h_{i,k}' = 0, \tag{11}$$
$$h_{j,l}'h_{i,k}' = 0, \tag{12}$$
$$h_{i,kl}'' = 0. \tag{13}$$

Eq. (11) indicates that $Z_i$ is a function of at most one of $\hat{Z}_k$ and $\hat{Z}_l$, while Eq. (12) implies that given that $Z_i$ and $Z_j$ are adjacent in Markov network $\mathcal{M}_Z$, at most one of them is a function of $\hat{Z}_k$ or $\hat{Z}_l$. $\qquad\square$

## B  PROOF OF THEOREM 1

First, we introduce the following lemma, which will be used in the proof.

**Lemma 3.** *For any invertible matrix A, there exists a permutation of its row such that the diagonal entries of the resulting matrix are nonzero.*

*Proof.* Suppose by contradiction that there exists at least a zero diagonal entry for every row permutation. By Leibniz formula, we have

$$\det(A) = \sum_{\sigma \in \mathcal{S}_n} \left( \text{sgn}(\sigma) \prod_{i=1}^n a_{\sigma(i),i} \right),$$

where $\mathcal{S}_n$ denotes the set of $n$-permutations. Since there exists a zero diagonal entry for every permutation, we have

$$\prod_{i=1}^n a_{\sigma(i),i} = 0, \quad \forall \sigma \in \mathcal{S}_n,$$

which implies $\det(A) = 0$ and that matrix $A$ is not invertible. This is a contradiciton with the assumption that $A$ is invertible. $\qquad\square$

We now provide the proof of Theorem 1

**Theorem 1** (Identifiability of Latent Markov Network). *Let the observations be sampled from the data generating process in Eq. (1), and $\mathcal{M}_Z$ be the Markov network over $Z$. Suppose that Assumptions A1 and A2 from Theorem 1 holds. Suppose also that we learn $(\hat{g}, \hat{f}, p_{\hat{Z}})$ to achieve Eq. (2) with the minimal number of edges of Markov network $\mathcal{M}_{\hat{Z}}$ over $\hat{Z}$. Then, the Markov network $\mathcal{M}_{\hat{Z}}$ over estimated hidden variables $\hat{Z}$ is isomorphic to the true latent Markov network $\mathcal{M}_Z$.*

*Proof.* Based on Lemma 3, there exists a permutation $\pi$ of the estimated hidden variables, denoted as $\hat{Z}_\pi$, such that

$$\frac{\partial Z_i}{\partial \hat{Z}_{\pi(i)}} \neq 0, \quad i = 1, \ldots, n. \tag{14}$$

Suppose that $Z_i$ and $Z_j$ are adjacent in the Markov network $\mathcal{M}_Z$ over $Z$, but $\tilde{Z}_{\pi(i)}$ and $\tilde{Z}_{\pi(i)}$ are not adjacent in the Markov network $\mathcal{M}_{\hat{Z}}$ over $\hat{Z}$. By Lemma 1, at most one of $Z_i$ and $Z_j$ is a function of $\tilde{Z}_{\pi(i)}$ and $\tilde{Z}_{\pi(i)}$. This is clearly a contradiction with Eq. (14).

Therefore, we have shown by contradiction that, if $Z_i$ and $Z_j$ are adjacent in the Markov network $\mathcal{M}_Z$ over $Z$, then $\tilde{Z}_{\pi(i)}$ and $\tilde{Z}_{\pi(i)}$ are adjacent in the Markov network $\mathcal{M}_{\hat{Z}_\pi}$ over variables $\hat{Z}_\pi$. That is, all edges in $\mathcal{M}_Z$ must be present in Markov network $\mathcal{M}_{\hat{Z}_\pi}$ over variables $\hat{Z}_\pi$. Also, note that the true model $(g, f, p_Z)$ is clearly one of the solutions that achieves Eq. (2). Thus, under sparsity constraint on the edges of $\mathcal{M}_{\hat{Z}}$, we conclude that the Markov network $\mathcal{M}_{\hat{Z}_\pi}$ over $\hat{Z}_\pi$ must be identical to the Markov network $\mathcal{M}_Z$ over $Z$, □

## C  PROOF OF THEOREM 2

We first state the following lemma that is used to prove Statement (b) of Theorem 2. The proof is a straightforward consequence of Cayley–Hamilton theorem and is omitted here.

**Lemma 4.** *Let $A$ be an $n \times n$ invertible matix. Then, it can be expressed as a linear combination of the powers of $A$, i.e.,*

$$A^{-1} = \sum_{k=0}^{n-1} c_k A^k$$

*for some appropriate choice of coefficients $c_0, c_1, \ldots, c_{n-1}$.*

Now consider the Markov network $\mathcal{M}_Z$ over hidden causal variables $Z$. Let $N_{Z_i}$ be the set of neighbors of $Z_i$ in $\mathcal{M}_Z$. We the following result that relates a matrix to its inverse, given that such matrix satisfies certain property defined by $\mathcal{M}_Z$.

**Proposition 1.** *Consider a Markov network $\mathcal{M}_Z$ over $Z$. Let $N_{Z_i}$ be the set of neighbors of $Z_i$ in $\mathcal{M}_Z$, and $A$ be an $n \times n$ invertible matrix. For each $i \neq j$ where $Z_j$ is not adjacent to some nodes in $\{Z_i\} \cup N_{Z_i}$, suppose $A_{ij} = 0$. Then, we have $A_{ij}^{-1} = 0$.*

*Proof.* By Lemma 4, $A^{-1}$ can be expressed as linear combination of the powers of $A$. Therefore, it suffices to prove that, each matrix power $A^k$ satisfies the following property: $A_{ij}^k = 0$ for each $i \neq j$ where $Z_j$ is not adjacent to some nodes in $\{Z_i\} \cup N_{Z_i}$. We proceed with mathematical induction on $k$. By definition, the property holds in the base case where $k = 1$.

Now suppose that the property holds for $A^k$. We prove by contradiction that the property holds for $A^{k+1}$. Suppose by contradiction that $A_{ij}^{k+1} \neq 0$ for some $i \neq j$ where $Z_j$ is not adjacent to some nodes in $\{Z_i\} \cup N_{Z_i}$. This implies that one of the following cases holds:

- Case (a): $Z_j$ is not adjacent to $Z_i$ in $\mathcal{M}_{G_Z}$.

- Case (b): There exists $Z_l \in N_{Z_i} \setminus \{Z_j\}$ such that $Z_j$ and $Z_l$ are not adjacent in $\mathcal{M}_{G_Z}$.

Since $A_{ij}^{k+1} = \sum_{r=0}^{n} A_{ir}^k A_{rj}$, the assumption $A_{ij}^{k+1} \neq 0$ implies that there must exist $m$ such that $A_{im}^k A_{mj} \neq 0$, i.e., $A_{im}^k \neq 0$ and $A_{mj} \neq 0$. Since both $A^k$ and $A$ satisfy the property, this indicates (i) $Z_m$ is adjacent to $Z_i$ and all nodes in $N_{Z_i} \setminus \{Z_m\}$, and (ii) $Z_j$ is adjacent to $Z_m$ and all nodes in $N_{Z_m} \setminus \{Z_j\}$. We consider the following cases:

- Case of $m = l$: By (ii), $Z_j$ is adjacent to $Z_l$, which contradicts Case (b) above. Also, we know that $Z_l$ is adjacent to $Z_i$ by (i), which indicates that $Z_i$ is adjacent to $Z_j$, contradicting Case (a) above.

- Case of $m \neq l$: By (i) and (ii), $Z_m$ is adjacent to $Z_i$ and $Z_j$ is adjacent to $Z_m$, implying that $Z_i$ and $Z_j$ are adjacent, which is contradictory with Case (a) above. Furthermore, since $Z_l$ is a neighbor of $Z_i$, we know that $Z_m$ and $Z_l$ are adjacent by (i). Also, by (ii), $Z_j$ is adjacent to $Z_l$, which contradicts Case (b) above.

In either of the cases above, there is a contradiction. $\qquad\square$

We are now ready to give the following result.

**Theorem 2** (Identifiability of Hidden Causal Variables). *Let the observations be sampled from the data generating process in Eq. (1), and $\mathcal{M}_Z$ be the Markov network over $Z$. Let $N_{Z_i}$ be the set of neighbors of variable $Z_i$ in $\mathcal{M}_Z$. Suppose that Assumptions A1 and A2 from Theorem 1 holds. Suppose also that we learn $(\hat{g}, \hat{f}, p_{\hat{Z}})$ to achieve Eq. (2) with the minimal number of edges of Markov network $\mathcal{M}_{\hat{Z}}$ over $\hat{Z}$. Then, there exists a permutation $\pi$ of the estimated hidden variables, denoted as $\hat{Z}_\pi$, such that the corresponding statements hold for $i = 1, \ldots, n$:*

*(a) $Z_i$ is a function of $\hat{Z}_{\pi(i)}$ and a (possibly empty) subset of the variables in $\{\hat{Z}_{\pi(j)} \mid Z_j$ is adjacent to $Z_i$ and all other neighbors of $Z_i$ in $\mathcal{M}_Z\}$.*

*(b) $\hat{Z}_{\pi(i)}$ is a function of $Z_i$ and a (possibly empty) subset of the variables in $\{Z_j \mid Z_j$ is adjacent to $Z_i$ and all other neighbors of $Z_i$ in $\mathcal{M}_Z\}$.*

*Proof.* We provide proof for both part of the statements.

**Proof of Statement (a):**

By Theorem 1 and its proof, there exists a permutation $\pi$ of the estimated variables, denoted as $\hat{Z}_\pi$, such that the Markov network $\mathcal{M}_{\hat{Z}_\pi}$ over $\hat{Z}_\pi$ is identical to $\mathcal{M}_Z$, and that

$$\frac{\partial Z_i}{\partial \hat{Z}_{\pi(i)}} \neq 0, \quad i = 1, \ldots, n.$$

Clearly, each variable $Z_i$ is a function of $\hat{Z}_{\pi(i)}$.

We first show that if $Z_j$ is not adjacent to $Z_i$ in $\mathcal{M}_Z$, then $Z_i$ cannot be a function of $\hat{Z}_{\pi(j)}$. Since $Z_i$ and $Z_j$ are not adjacent in $\mathcal{M}_Z$, we know that $\hat{Z}_{\pi(i)}$ and $\hat{Z}_{\pi(j)}$ are not adjacent in $\mathcal{M}_{\hat{Z}_\pi}$. By Lemma 1, $Z_i$ is a function of at most one of $\hat{Z}_{\pi(i)}$ and $\hat{Z}_{\pi(j)}$, which implies that $Z_i$ cannot be a function of $\hat{Z}_{\pi(j)}$, because we have shown that $Z_i$ is a function of $\hat{Z}_{\pi(i)}$.

To refine further, now suppose that $Z_j$ is adjacent to $Z_i$, but not adjacent to some $Z_k \in N_{Z_i} \setminus \{Z_j\}$. Since $\mathcal{M}_Z$ and $\mathcal{M}_{\hat{Z}_\pi}$ are identical, $\hat{Z}_{\pi(j)}$ is also not adjacent to $\hat{Z}_{\pi(k)}$ in $\mathcal{M}_{\hat{Z}_\pi}$. Since $Z_i$ and $Z_k$ are adjacent in $\mathcal{M}_Z$, by Lemma 1, at most one of them is a function of $\hat{Z}_{\pi(j)}$ or $\hat{Z}_{\pi(k)}$. This implies that $Z_i$ cannot be a function of $\hat{Z}_{\pi(j)}$, because we have shown that $Z_k$ is a function of $\hat{Z}_{\pi(k)}$.

**Proof of Statement (b):**

By Statement (a), we know that $Z_i$ is a function of at most the variables in $\hat{Z}_{\pi(i)} \cup \{\hat{Z}_{\pi(j)} \mid Z_j$ is adjacent to $Z_i$ and all other neighbors of $Z_i$ in $\mathcal{M}_Z\}$. Therefore, for each $i \neq j$ where $Z_j$ is not adjacent to some nodes in $\{Z_i\} \cup N_{Z_i}$, we have

$$\frac{\partial Z_i}{\partial \hat{Z}_{\pi(j)}} = 0.$$

By Proposition 1, we have

$$\left( \frac{\partial Z}{\partial \hat{Z}_\pi} \right)^{-1}_{ij} = 0,$$

which, by inverse function theorem, implies

$$\frac{\partial \hat{Z}_{\pi(j)}}{\partial Z_i} = \left( \frac{\partial Z}{\partial \hat{Z}_\pi} \right)^{-1}_{ij} = 0.$$

This implies that $Z_i$ cannot be a function of $\hat{Z}_{\pi(j)}$. $\qquad\square$

## D    PROOF OF LEMMA 2 AND THEOREM 3

**Lemma 2.** *Given a latent causal graph $\mathcal{G}_Z$ and distribution $P_Z$ with its Markov Network $\mathcal{M}_Z$, under Markov assumption, the undirected graph defined by $\mathcal{M}_Z$ is a subgraph of the moralized graph of the true causal DAG $G$.*

*Proof.* Let $Z_j$ and $Z_k$, $j \neq k$ be two variables that $i$ and $j$ are not adjacent in the moralized graph of $\mathcal{G}_Z$. Then it suffices to show that $(j, k) \notin \mathcal{E}(\mathcal{M}_Z)$ and $(k, j) \notin \mathcal{E}(\mathcal{M}_Z)$. Because they are not adjacent in the moralized graph of $\mathcal{G}_Z$, they must not be adjacent in $\mathcal{G}_Z$ and must not share a common child in $\mathcal{G}_Z$. Thus, $j$ and $k$ are d-separated conditioning on $V \setminus \{j, k\}$, which implies the conditional independence $Z_j \perp\!\!\!\perp Z_k | Z \setminus \{Z_j, Z_k\}$ based on the Markov assumption on $\{\mathcal{G}_Z, P_Z\}$. Then we have $(j, k) \notin \mathcal{E}(\mathcal{M}_Z)$ and $(k, j) \notin \mathcal{E}(\mathcal{M}_Z)$. $\square$

**Theorem 3.** *Given a causal DAG $\mathcal{G}_Z$ and distribution $P_Z$ with its Markov Network $\mathcal{M}_Z$, under Markov assumption, the undirected graph defined by $\mathcal{M}_Z$ is the moralized graph of the true causal DAG $\mathcal{G}_Z$ if and only if the SAF and SUCF assumptions are satisfied.*

*Proof.* We prove both directions as follows.

**Sufficient condition.**    We prove it by contradiction. Suppose that the structure defined by $\text{supp}(\mathcal{M}_Z)$ is not equivalent to the moralized graph of $\mathcal{G}_Z$. Then, according to Lem. 2, there exists a pair of variables $Z_j$ and $Z_k$, $j \neq k$ that $i$ and $j$ are adjacent in the moralized graph but $(j, k) \notin \mathcal{E}(\mathcal{M}_Z)$ and $(k, j) \notin \mathcal{E}(\mathcal{M}_Z)$. Thus, we have $Z_j \perp\!\!\!\perp Z_k | Z \setminus \{Z_j, Z_k\}$. Then we consider the following two cases:

- If variables $Z_j$ and $Z_k$ correspond to a pair of neighbors in $\mathcal{G}_Z$, then they are adjacent. Together with the conditional independence relation $Z_j \perp\!\!\!\perp Z_k | Z \setminus \{Z_j, Z_k\}$, this implies that the SAF assumption is violated.

- If variables $Z_j$ and $Z_k$ correspond to a pair of non-adjacent spouses in $\mathcal{G}_Z$. Then they have an unshielded collider, indicating that the SUCF assumption is violated.

**Necessary condition.**    We prove it by contradiction. Suppose SUCF or SAF is violated, we have the following two cases:

- Suppose SUCF is violated, i.e., there exists an unshielded collider $j \to i \leftarrow k$ in the DAG $\mathcal{G}_Z$ such that $Z_j \perp\!\!\!\perp Z_k | Z \setminus \{Z_j, Z_k\}$. This conditional independence relation indicates that $(j, k) \notin \mathcal{E}(\mathcal{M}_Z)$ and $(k, j) \notin \mathcal{E}(\mathcal{M}_Z)$. Since $j$ and $k$ are spouses, there exists an edge between them in the moralized graph of $\mathcal{G}_Z$, but is not contained in the structure defined by $\mathcal{M}_Z$, showing that they are not the same.

- Or, suppose SAF is violated, i.e., there exists a pair of neighbors $j$ and $k$ in the DAG $\mathcal{G}_Z$ such that $Z_j \perp\!\!\!\perp Z_k | Z \setminus \{Z_j, Z_k\}$. This conditional independence relation indicates that $(j, k) \notin \mathcal{E}(\mathcal{M}_Z)$ and $(k, j) \notin \mathcal{E}(\mathcal{M}_Z)$. Because $j$ and $k$ are adjacent in $\mathcal{G}_Z$, clearly they are also adjacent in the moralized graph of $\mathcal{G}_Z$. However, the edge between them is not contained in the structure defined by $\mathcal{M}_Z$, showing that they are not the same.

Thus, when SUCF or SAF is violated, the structure defined by $\mathcal{M}_Z$ is the moralized graph of the true DAG $\mathcal{G}_Z$. $\square$

## E    PROOF OF THEOREM 4

**Theorem 4** (Identifiability of Latent Factors)**.** *Let the observations be sampled from the data generating process in Eq. (1), and $\mathcal{M}_Z$ be the Markov network over $Z$. Suppose that the following assumptions hold:*

- *A1 (Smooth and positive density): The probability density function of latent causal variables is smooth and positive, i.e. $p_Z$ is smooth and $p_Z > 0$ over $\mathbb{R}^n$.*

- *A2 (Modular changes): $\theta_i$ across different $i$ are not related via equality constraints.*

- *A3 (Sufficient changes): for $Z \in \mathbb{R}^n$ and each value of $\theta$, the vectors $w(\theta)$ are linearly independent, where $w(\theta)$ is defined as*

$$w(\theta) = \left( \frac{\partial \log p(Z_1 \mid PA(Z_1); \theta_1)}{\partial \theta_1}, \ldots, \frac{\partial \log p(Z_n \mid PA(Z_n); \theta_n)}{\partial \theta_n}, \right.$$
$$\left. \frac{\partial^2 \log p(Z_1 \mid PA(Z_1); \theta_1)}{\partial \theta_1^2}, \ldots, \frac{\partial^2 \log p(Z_n \mid PA(Z_n); \theta_n)}{\partial \theta_n^2} \right).$$

*Suppose that we learn $(\hat{g}, \hat{f}, p_{\hat{Z}})$ to achieve Eq. (2). Then, $\theta_i$ are identifiable up to component-wise invertible transformation from the observations $X$ in multiple domains.*

*Proof.* Following the derivation in the proof of Lemma 1, we obtain

$$\log p(\hat{Z}; \hat{\theta}) = \log p(Z; \theta) - \log |J_q|,$$

which can be rewritten as

$$\sum_{i=1}^{d} \log p(\hat{Z} \mid PA(\hat{Z}_i); \hat{\theta}_i) = \sum_{i=1}^{d} \log p(Z_i \mid PA(Z_i); \theta_i) - \log |J_q|. \tag{15}$$

Let $\hat{\eta}_i^{(k)}(\hat{\theta}_i) := \log p(\hat{Z} \mid PA(\hat{Z}_i); \hat{\theta}_i)$ and $\eta_i(\theta_i) := \log p(Z_i \mid PA(Z_i); \theta_i)$. We have

$$\sum_{i=1}^{d} \hat{\eta}_i(\hat{\theta}_i) = \sum_{i=1}^{d} \eta_i(\theta_i) - \log |J_h|. \tag{16}$$

With Assumption A2, the second-order cross derivative of Eq. (16) w.r.t. $\hat{\theta}_i$ and $\hat{\theta}_j$ is

$$0 = \sum_{l=1}^{d} \left( \eta_l''(\theta_l) \frac{\partial \theta_l}{\partial \hat{\theta}_i} \frac{\partial \theta_l}{\partial \hat{\theta}_j} + \eta_l'(\theta_l) \frac{\partial^2 \theta_l}{\partial \hat{\theta}_i \partial \hat{\theta}_j} \right).$$

If for each value of $\theta$, $\eta_1''(\theta_1), \eta_1'(\theta_1), \eta_2''(\theta_2), \eta_2'(\theta_2), \ldots, \eta_d''(\theta_d), \eta_d'(\theta_d)$ are linearly independent, then $\frac{\partial \theta_l}{\partial \hat{\theta}_i} \frac{\partial \theta_l}{\partial \hat{\theta}_j} = 0$ and $\frac{\partial^2 \theta_l}{\partial \hat{\theta}_i \partial \hat{\theta}_j} = 0$.

The statement $\frac{\partial \theta_l}{\partial \hat{\theta}_i} \frac{\partial \theta_l}{\partial \hat{\theta}_j} = 0$ implies that each $\theta_l$ is a function of at most one of $\hat{\theta}_1, \ldots, \hat{\theta}_n$. By considering the inverse of Jacobian matrix, we also have $\frac{\partial \hat{\theta}_l}{\partial \theta_i} \frac{\partial \hat{\theta}_l}{\partial \theta_j} = 0$, indicating that each $\hat{\theta}_l$ is a function of at most one of $\theta_1, \ldots, \theta_n$. Combining both of them, $\theta_i$ are identifiable up to component-wise invertible transformation. $\square$

