# OpenReview forum: "Revealing Hidden Causal Variables and Latent Factors from Multiple Distributions"
_ICLR.cc/2024/Conference — Submitted to ICLR 2024_

### Official Review · Reviewer_d7mo · 2023-11-01

**Soundness:** 2 fair
**Presentation:** 3 good
**Contribution:** 2 fair
**Rating:** 3
**Confidence:** 3

**Summary:**

In this paper, causal inference with hidden factors is investigated. The model studied in equation (1) is quite general, and the focus of the work is to understand when and how the causal structure can be estimated from observations.

In the first set of results, the authors determine conditions under which the Markov graph underlying the conditional dependency structure of the variables can be uniquely identified from the distribution of observations (Thms 1 - 3). Furthermore, the latent factors which affect the hidden variables can also be identified (Thm 5). Finally, the authors assess their methods through simulations.

**Strengths:**

The authors study a very general model, and establish fundamental conditions for the identifiability of causal structures and latent factors. The fundamental contributions that are made for this model therefore have important and widespread applications. The paper is clearly written.

**Weaknesses:**

While the bulk of the paper establishes fundamental results for causal inference, this seems to have little impact on the design or guarantees of machine learning algorithms. Though there is a Simulations section in which the authors apply their theory, there seem to be many details and analysis missing. For instance, it is not clear to me how the data is generated, and the results shown in Figures 2 and 3 have little explanation. Hence it is quite challenging to understand what exactly the authors are trying to demonstrate, and how well their procedure performs.

The technical side seems reasonable. However, I didn't understand the significance of the linearly independent vectors (see, e.g., bullet point 2 of Thm 1). Please explain its importance.

**Questions:**

- On page 3, should it read $Z_j \to X_i$ if and only if $Zj \in PA(X_i)$?
- How should we imagine that $(\hat{g}, \hat{f}, p_{\hat{Z}})$ is learned to achieve Eq. (2)?
- Unclear how well structure is recovered, just from looking at Figure 2. What do these scatter plots represent? Same comment for Figure 3.
- In the text after Eqn (6), you state that $C^{(u)}$ and $S^{(u)}$ are a matrix and vector, respectively. Then in Eqn (7) you "divide" by $S^{(u)}$ in the right hand side. Could you elaborate on what this means

---

> ### Author Response · Authors · 2023-11-21
> **Detailed Responses to Your Comments (1/2)**
>
> We sincerely thank the reviewer for the time dedicated to reviewing our paper and the constructive suggestions. Our responses to these comments are given below.
>
> Q1: "While the bulk of the paper establishes fundamental results for causal inference, this seems to have little impact on the design or guarantees of machine learning algorithms."
>
> A1: Thanks for this insightful comment. The theoretical results have two major impacts on the design of machine learning algorithms: (1) learning  $(\hat{g}, \hat{f},p_{\hat{Z}})$ to achieve Eq. (2), and (2) applying sparsity constraint on the edges of Markov network over $\hat{Z}$. We will make this clear in the revised manuscript.
>
> Q2: "it is not clear to me how the data is generated, and the results shown in Figures 2 and 3 have little explanation. Hence it is quite challenging to understand what exactly the authors are trying to demonstrate, and how well their procedure performs."
>
> A2: Thanks for pointing this out. We assume that the influences from domain and its parents are location and scaling transformations. Specificically, for each component $z_i$ in each domain, suppose the parents are $PA(i)$, the $z_i=\sum_{j\in PA(i)} s_{j,i}z_j+b_i$, where $s_{j,i}$ are scalars sampled from $Unif[0.5, 2]$ and $b_i\in Unif[-2,2]$. We use 100 domains such that the number of environments are sufficient. As for the nonlinear mixing function, we follow the standard nonlinear ICA literature and construct a MLP whose output dimension is the same as input dimension. Each layer of MLP consists of a orthogonal matrix weight and a leaky-relu function. Since the weights are orthogonal and the activation is invertible, the MLP is invertible too. For the Y-structure, we have $z_1\rightarrow z_3 \leftarrow z_2$ and $z_3 \rightarrow z_4$. As shown in the Fig. 3(a)(b), the estimated $\hat{z}_1$ recovers $z_2$ and $\hat{z}_1$ is independent from $z_1$; the estimated $\hat{z}_2$ recovers $z_1$ and $\hat{z}_2$ is independent from $z_2$. Therefore, our estimated $\hat{z}_1, \hat{z}_2$ recovers the $z_2, z_1$ and their independence relationship. As for $\hat{z}_3$, it is dependent on $\hat{z}_1$ and $\hat{z}_2$. Therefore, we have $\hat{z}_1\rightarrow \hat{z}_3 \leftarrow \hat{z}_2 \Leftrightarrow z_2 \rightarrow z_3 \leftarrow z_1$. As for $\hat{z}_4$, it is dependent on all previous variables, i.e., $\hat{z}_1, \hat{z}_2, \hat{z}_3$. Since we have the sparisty regularization, we can remove the redundant link from $\hat{z}_1$ and $\hat{z}_2$ to $\hat{z}_4$. Therefore, we are able to recover the true latent cause structure. In addition, in eq 6, we have an adjacency matrix to learn, the learned adjacency matrix is identical to the true adjacency matrix, which further supports our theory.
>
> Q3: "I didn't understand the significance of the linearly independent vectors (see, e.g., bullet point 2 of Thm 1). Please explain its importance."
>
> A3: The linearly independent vectors are crucial to the proof of the theoretical result, and is commonly used in the literature [1]. Specifically, it allows us to obtain Eqs. (11), (12), and (13) from the equation above them that involves linear combinations of the corresponding terms. Intuitively speaking, it requires the distribution to vary sufficiently across different domains. We will explain this in the revised manuscript.
>
> Q4: "On page 3, should it read $Z_j \to X_i$ if and only if $Z_j \in PA(X_i)$?"
>
> A4: Thanks for spotting this. It should read $Z_j\rightarrow Z_i$ if and only if $Z_j\in\textrm{PA}(Z_i)$, and we have fixed this typo in the revision.
>
> Q5: "How should we imagine that $(\hat{g}, \hat{f}, p_{\hat{Z}})$ is learned to achieve Eq. (2)?"
>
> A5: Thanks for this insightful question. We estimate a model $(\hat{g}, \hat{f},p_{\hat{Z}})$ that assumes the same data generating process as in Eq. (1) and matches the true distribution of $X$. We did not specify exactly how to achieve it, and leave the door open for different approaches to be used, such as normalizing flow or variational approaches. For example, we adopt a variational approach in Section 5. We have included this discussion in the updated manuscript.

---

> > ### Author Response · Authors · 2023-11-21
> > **Detailed Responses to Your Comments (2/2)**
> >
> > Q6: "Unclear how well structure is recovered, just from looking at Figure 2. What do these scatter plots represent? Same comment for Figure 3."
> >
> > A6: Thanks for your question. We assume that there are 4 components in $z$. In each subfigure, there are $x\times 4$ graphs, graph $i,j$ denotes the scatter plot between the true component $z_i$ and the estimated $z_j$. Firstly, we observe that there is one-to-one correpondence between the estimated components and the true components, supporting our component-wise identifiability theory. On the other hand, for the Y-structure, we have $z_1\rightarrow z3 \leftarrow z2$ and $z_3 \rightarrow z_4$. As shown in the Fig. 3(a)(b), the estimated $\hat{z}_1$ recovers $z_2$ and $\hat{z}_1$ is independent from $z_1$; the estimated $\hat{z}_2$ recovers $z_1$ and $\hat{z}_2$ is independent from $z_2$. Therefore, our estimated $\hat{z}_1, \hat{z}_2$ recovers the $z_2, z_1$ and their independence relationship. As for $\hat{z3}$, it is dependent on $\hat{z}_1$ and $\hat{z}_2$. Therefore, we have $\hat{z}_1\rightarrow \hat{z}_3 \leftarrow \hat{z}_2 \Leftrightarrow z_2 \rightarrow z_3 \leftarrow z_1$. As for $\hat{z}_4$, it is dependent on all previous variables, i.e., $\hat{z}_1, \hat{z}_2, \hat{z}_3$. Since we have the sparisty regularization, we can remove the redundant link from $\hat{z}_1$ and $\hat{z}_2$ to $\hat{z}_4$. Therefore, we are able to recover the true latent cause structure. In addition, in eq 6, we have an adjacency matrix to learn, the learned adjacency matrix is identical to the true adjacency matrix, which further supports our theory.
> >
> > Q7: "In the text after Eqn (6), you state that $C^{(u)}$ and $S^{(u)}$ are a matrix and vector, respectively. Then in Eqn (7) you "divide" by $S^{(u)}$ in the right hand side. Could you elaborate on what this means"
> >
> > A7: Sure, no problem! Yes, $S^{u}$ is a vector representing the domain-specific scaling parameter. In eq7, the output $\hat{\epsilon}$ is also a vector. It means that we perform element-wise division for each component.
> >
> > References:
> >
> > [1] Hyvärinen, Aapo, Ilyes Khemakhem, and Hiroshi Morioka. "Nonlinear independent component analysis for principled disentanglement in unsupervised deep learning." Patterns 2023.

---

### Official Review · Reviewer_xPxb · 2023-11-01

**Soundness:** 3 good
**Presentation:** 3 good
**Contribution:** 2 fair
**Rating:** 5
**Confidence:** 4

**Summary:**

This paper studies latent causal identification under multiple distributions where multiple distributions are modeled as mechanism change. The first part tries to latents under sparsity constraint using observational distribution only. The second part uses orthogonal changes to show that latent factors can be learned with multiple distributions.

**Strengths:**

The paper is well-written. It is quite easy to follow.

**Weaknesses:**

- Is the condition (sufficient change) of theorem 1 too strict? What are some practical use cases for this?

- The experiment section is limited, only on simulated data and very limited graphs.

- The number of latents needs to be known in advance.

- I am a little confused about one of the main objectives of the paper: What does identifying potential shifts even mean? Even if I can learn $\theta$ up to component-wise invertible transformations? What does knowing $\theta$ tell me? And I don’t think theorem 5 is verified by experiments either.
 - By the way, Can you explain how the last line of the proof of theorem 5 leads to the conclusion? Also where in the proof do you need the modular change assumption?

**Questions:**

- the sparsity condition is not made explicit in theorem 2? Can you state it formally?

- not sure how to interpret the results of theorem 1? it is not intuitive? If the main purpose of theorem 1 is to prove later theorem, maybe it should be a lemma?

-  The proof of theorem 1 uses assumption 3 which is not assumed by theorem 1?

- what's the relation between modular change and ICM? why do you use the "not related via equality constraint"? What if $\theta_i$ and $\theta_j$ are related by some inequalities? Does you result still hold?

- How is your multi-distribution setting different from soft interventions?

---

> ### Author Response · Authors · 2023-11-21
> **Detailed Responses to Your Comments (1/2)**
>
> We greatly appreciate the reviewer's constructive comments. We have tried to address all the concerns in the following.
>
> Q1: "Is the condition (sufficient change) of theorem 1 too strict? What are some practical use cases for this?"
>
> A1: Thanks for this question. The requirement of a sufficient number of environments is actually standard in the literature (see e.g. a recent survey [1]). Various real-world experimental results (e.g., visual disentanglement [2], domain adaptation [3], video analysis [4], and image-to-image translation [5]) in this line of research indicate that it is likely to hold in practice. Meanwhile, for many tasks, the number of environments is actually accessible, just like the number of domains in transfer learning and the number of time indices in time series data. We have updated the manuscript to include this discussion.
>
> Q2: "The experiment section is limited, only on simulated data and very limited graphs."
>
> A2: Thanks for pointing this out. Since the main focus of our manuscript is the theoretical identifiability results, our experiments mainly serve to validate our theory. The figure 3 shows that we are able to recover the true components and true causal strucuture when the components are causally related instead of mutually independent.
>
> Q3: "The number of latents needs to be known in advance."
>
> A3: Thanks for this comment. Although it is a rather standard assumption in causal representation learning to assume that the number of latents is known in advance (e.g., [1, 6]), let us discuss how to deal with number of latent variables in practice. Some possible approaches are (1) selecting the number of latent variables via model selection, or (2) use a mask to automatically select the number of latent variables, similar to [7]. We will include this discussion in the revised manuscript.
>
> Q4: "What does identifying potential shifts even mean? Even if I can learn  $\theta$  up to component-wise invertible transformations? What does knowing  $\theta$ tell me? And I don’t think theorem 5 is verified by experiments either."
>
> A4: $\theta$ denotes the latent (changing) factor (or effective parameters) associated with each latent structural equation model, which governs the changes in  causal mechanisms. Knowing $\theta$ tells us whether a particular causal mechanism changes across certain domains.
>
> Q5: "Can you explain how the last line of the proof of theorem 5 leads to the conclusion? Also where in the proof do you need the modular change assumption?"
>
> A5: Thanks a lot for this question. The statement $\frac{\partial \theta_l}{\partial \hat{\theta}_i} \frac{\partial \theta_l}{\partial \hat{\theta}_j} = 0$ implies that each $\theta_l$ is a function of at most one of $\hat{\theta}_1,\dots,\hat{\theta}_n$. By considering the inverse of Jacobian matrix, we also have $\frac{\partial \hat{\theta}_l}{\partial \theta_i} \frac{\partial \hat{\theta}_l}{\partial \theta_j} = 0$, indicating that each $\hat{\theta}_l$ is a function of at most one of $\theta_1,\dots,\theta_n$. Combining both of them, $\theta_i$ are identifiable up to component-wise invertible transformation.
>
> Furthermore, the modular change assumption is needed to obtain the equation below Eq. (16). We have modified the manuscript to make these clear.
>
> Q6: "the sparsity condition is not made explicit in theorem 2? Can you state it formally?"
>
> A6: The sparsity constraint implies the minimal number of edges of Markov network $\mathcal{M}_{\hat{Z}}$ over $\hat{Z}$. We have upadated the manuscript to make this explicit.
>
> Q7: "not sure how to interpret the results of theorem 1? it is not intuitive? If the main purpose of theorem 1 is to prove later theorem, maybe it should be a lemma?"
>
> A7: Thanks for the constructive suggestion. We have modified Theorem 1 to be a lemma in the revised manuscript.
>
> Q8: "The proof of theorem 1 uses assumption 3 which is not assumed by theorem 1?"
>
> A8: Thanks for your careful reading and spotting this. We have fixed the typo in the proof of Theorem 1. That is, the proof does not involve Assumption A3, which instead should be Assumption A2.

---

> ### Author Response · Authors · 2023-11-21
> **Detailed Responses to Your Comments (2/2)**
>
> Q9: "what's the relation between modular change and ICM? why do you use the "not related via equality constraint"? What if $\theta_i$ and $\theta_j$ are related by some inequalities? Does you result still hold?"
>
> A9: Thanks for asking this question. One can consider the modular change condition as an instantiation of ICM. We will discuss this conenction in the revised manuscript to explain clearly the relation between modular changes and ICM.
>
> In our case, we have not been aware the situation where $\theta_i$ and $\theta_j$ are related by some inequalities. Intuitively, if it is possible for them to be related by inequality cosntraints, then in this case, one should include inequality constraints into Assumption A2.
>
> Q10: "How is your multi-distribution setting different from soft interventions?"
>
> A10: If soft intervention is defined this way as in [8], then it is completely general, as long as there is change in the conditional distribution. If so, then we assume soft intervention for all latent causal variables. We will discuss this in the revised manuscript.
>
> References:
>
> [1] Hyvärinen, Aapo, Ilyes Khemakhem, and Hiroshi Morioka. "Nonlinear independent component analysis for principled disentanglement in unsupervised deep learning." Patterns 2023.
>
> [2] Khemakhem, Ilyes, et al. "Ice-beem: Identifiable conditional energy-based deep models based on nonlinear ica." NeurIPS 2020
>
> [3] Kong, Lingjing, et al. "Partial Identifiability for Domain Adaptation." ICML 2022
>
> [4] Yao, Weiran, et al. "Learning temporally causal latent processes from general temporal data." ICLR 2022
>
> [5] Xie, Shaoan, et al. "Unpaired Image-to-Image Translation With Shortest Path Regularization." CVPR 2023.
>
> [6] von Kügelgen, Julius, et al. "Nonparametric Identifiability of Causal Representations from Unknown Interventions." NeurIPS 2023
>
> [7] Xie, Shaoan, et al. "Multi-domain image generation and translation with identifiability guarantees." ICLR 2022.
>
> [8] Eberhardt, F., et al. "Direct Causes and the Trouble with Soft Interventions." Erkenntnis 2014.

---

### Official Review · Reviewer_8efN · 2023-11-02

**Soundness:** 1 poor
**Presentation:** 2 fair
**Contribution:** 2 fair
**Rating:** 3
**Confidence:** 3

**Summary:**

This paper studies identifiability questions for causal latent systems, in a situation where observations from multiple and diverse environments are available. The diversity here is a necessary condition for the method.

It is shown that one can use a certain known characterisation of conditional independence using derivatives of the density to partially identify the dependence structure of the latent system, provided it is observed in a diverse enough number of environments.


Experiments on small synthetic examples are performed to demostrate the general problem setup and identifiability in it.

**Strengths:**

The subject of identifiability in causal systems is important.
I have not seen the characterisation of conditional independence via density derivatives used in recent literature, and believe it is of interest.

**Weaknesses:**

First, there are several critical issues with presentation. Second, the setup considered in the paper is quite restrictive. In addition, the main "sufficient changes" assumpion is not discussed. It is not discussed when we can expect it to hold, nor is it evaluated in any actual scenarios. This is the main issue.  Third, the experiments are performed on small toy examples, and use known architectures of causal discovery.  It is not clear what is their added value.

In more detail:

**Presentation:**
**(a)** the presentation is missleading as it positions the paper as "treating the case of multiple environments, which is not treated in the litarture". This makes the impression that the approach is a generalisation of a single environment case. However, it is not. The proposed methods can not be used for a single environment. In a way, the method exploits the differences in environments, and there must be many of them.   **(b)** Some critical definitions are not given in the main text. Specifically, "Modular Changes" of Theorem 5 is not defined, and "sparsity constraint" of Theorem 3.  **(c)** Literature: there is a significant amount of research on multiple environments via causality that is not mentioned in the paper. The NN architectures presented in Experiments are small modifications of known arhcitectures (normalising flows flows for causal discovery).  Proper references should be given.



**Problem Setting and The Sufficient Changes Assumption**:
The setting (1) is restrictive in two ways: First, the observations $X$ are a deterministic function, i.e not noisy observations. Second, and more importantly, the observation map $g$ is assumed to be _invertible_. For at least somewhat smooth maps this means dimension of $X$ and $Z$ must be the same, which is unrealistic. How important is invertability?

The fundamental assumption of this paper, of Sufficient Changes, appears very strong. It requires independence for every point $z$. The number of environments must be at least as large as the number of variables (if same set of environments is good for all $z$). This condition must be discussed. When does it hold? Examples? I would suggest removing the Experiments section, and discussing this condition in detail.



**Experiments**:
In addition to what mentioned on the experiments above:
Regarding the following phrase in the end of Section 5:
>However, when the noises are Gaussian, it becomes more challenging and we observe that some components are mixed, which further aligns with our theory (e.g., Theorem 3) and demonstrate the benefit of using suitable parameterization.

I do not see how this results aligns with the theory, and in particular with Theorem 3. Theorem 3 does not appear to have assumptions that distinguish between Gaussainity and non Gaussianity. Please exlain.

**Questions:**

Please see above.

---

> ### Author Response · Authors · 2023-11-21
> **Detailed Responses to Your Comments (1/2)**
>
> Thanks for your time and effort in reviewing our manuscript. Below we give a point-by-point response to the comments.
>
> Q1: The presentation might be misleading since it makes the impression that the approach is a generalisation of a single environment case.
>
> A1: Thanks for your question. We fully agree with you that our method exploits the distributional change to identify latent causal variables and the structure among them. Actually, we thought it is exactly the considered task and didn’t treated it as a generalisation of the single environment case. In fact, the single environment case is way more difficult and most previous works in causal representation learning assume a sufficient number of different environments (e.g., [1,2]). Therefore, the multi-environments case is not a generalization of a single environment case. We have modified the introduction in order to avoid any potential confusion.
>
> Q2: Lack of some definitions.
>
> A2: Thank you for the reminder, and apologize for the potential confusion. The term “modular changes” is defined in A3 of Theorem 5 (“$\theta_i$ across different $i$ are not related via equality constraints”). The sparsity constraint implies the minimal number of edges of Markov network $\mathcal{M}_{\hat{Z}}$ over $\hat{Z}$.  We have updated the theorem statement to make this explicit.
>
> Q3: Related work on multiple environments via causality and nomalizing flows for causal discovery.
>
> A3: Thanks for the suggestion. As detailed in A1, our theoretical results take the existence of multiple environments as an assumption of the identifiability theory for causal representation learning, which differs from previous works trying to address the challenge of multiple environments (e.g., CD-NOD [3]). And one can consider causal representation learning as an extension of causal discovery, which aims at identifying both the latent causal variables and hidden structures. We have incorporated the discussion and futher highlighted the differences in the manuscript as well as the recommended references.
>
> Q4: Observations are assumed to be noiseless and the map is invertible.
>
> A4: We appreciate the insightful questions. Let us apologize for the typo in the paragraph before Eq. 9. The map is actually injective but not invertible, which is a standard assumption in the previous work in causal representation learning [2,4]. Moreover, since most identifiability theorems in the literature stems from the theory of nonlinear ICA, many works follows the setting of it, where the observation $X$ is a deterministic (nonlinear) function of latent variables $Z$. Our work lies in this line of research and thus follows the previous setting. We will provide additional discussion on it to make the context clearer, according to your constructive suggestions.
>
> Q5: More discussion on the assumption of sufficient changes.
>
> A5: Thanks a lot for your great suggestions. In light of these, we have added more discussions in the updated manuscript, including more real-world examples. At the same time, we would like to mention that this assumption is common in the literature. For example, [4, 5] require $2n+1$ environments in order for the system to be identifiable, where $n$ is the number of variables. Some real-world experiments (e.g., visual disentanglement [6], domain adaptation [7], video analysis [8], and image-to-image translation [9]) also suggests that this assumption is likely to hold at least in certain real-world scenarios.

---

> > ### Author Response · Authors · 2023-11-21
> > **Detailed Responses to Your Comments (2/2)**
> >
> > Q6: Why Gaussianity has been mentioned in experiments but Theorem 3 does not distinguish between Gaussianity and non-Gaussianity?
> >
> > A6: Thanks for raising this point. Actually, Theorem 3 perserves some degree of mixture after our estimation, and that is why we stated that the observation that some components are mixed together align with Theorem 3. Intuitively, we conjecture that non-Gaussianity might be helpful for further decreasing the indeterminacy, just as that in the linear ICA case. Thus, we also tested our model on dataset with non-Gaussianity, and our empirical results indeed show that introducing non-Gaussianity is helpful. However, this is just an empirical validation of our intuition, so we only mentioned it in experiments.
> >
> > References:
> >
> > [1] von Kügelgen, Julius, et al. "Nonparametric Identifiability of Causal Representations from Unknown Interventions." NeurIPS 2023
> >
> > [2] Hyvärinen, Aapo, Ilyes Khemakhem, and Hiroshi Morioka. "Nonlinear independent component analysis for principled disentanglement in unsupervised deep learning." Patterns 2023.
> >
> > [3] Huang, Biwei, et al. "Causal discovery from heterogeneous/nonstationary data." JMLR 2020.
> >
> > [4] Khemakhem, Ilyes, et al. "Variational autoencoders and nonlinear ica: A unifying framework." ATSTATS 2020
> >
> > [5] Hyvarinen, Aapo, Hiroaki Sasaki, and Richard Turner. "Nonlinear ICA using auxiliary variables and generalized contrastive learning." AISTATS 2019
> >
> > [6] Khemakhem, Ilyes, et al. "Ice-beem: Identifiable conditional energy-based deep models based on nonlinear ica." NeurIPS 2020
> >
> > [7] Kong, Lingjing, et al. "Partial Identifiability for Domain Adaptation." ICML 2022
> >
> > [8] Yao, Weiran, et al. "Learning temporally causal latent processes from general temporal data." ICLR 2022
> >
> > [9] Xie, Shaoan, et al. "Unpaired Image-to-Image Translation With Shortest Path Regularization." CVPR. 2023.

---

### Official Review · Reviewer_mmWQ · 2023-11-03

**Soundness:** 2 fair
**Presentation:** 1 poor
**Contribution:** 1 poor
**Rating:** 3
**Confidence:** 2

**Summary:**

The paper works on the problem of recovering the hidden causal structure of latent variables given observational data from different environments. Authors show, that under a sparsity constraint of an underlying Markov network and a sufficient number of changes across environments, the Markov network can be recovered up to the simple indeterminacies. Furthermore, under additional mild assumptions, the recovered Markov network coincides with the moralized causal DAG. Additionally, latent factors can be recovered too, under similar assumptions. The authors provide two different implementations following the theory and do several synthetic experiments.

**Strengths:**

The problem of identifying a latent structure is highly relevant in causal inference research. To tackle it, the authors made some interesting theoretical work by applying the proof technique from Lin (1997) based on second-order derivatives of the log-density. Also, I found the connection between Markov networks and causal DAGs interesting.  Additionally, I appreciate the attempts to make the proposed approach as general as possible.

**Weaknesses:**

I identified several important flaws, which need to be addressed by the authors. Those include connections to previous works, lack of rigour and questionable correctness of the statements, realisticness of the assumptions, and experimental evidence/evaluation issues.

**Connections to previous works**. It seems like the identification technique with second-order derivatives of the log-density from  Lin (1997) is a standard technique for identifying structural causal models (SCMs), e.g., used in [1, 2]. Also, I don’t understand the connections between existing identifiable SCMs, like additive noise models and post-non-linear models, and the statements of Theorems 1 and 3, as those SCMs must be special cases of the proposed Theorems.

**Lack of rigour & questionable correctness**. I found it hard to understand the claims of the Theorems, as some notation was not introduced properly or sufficiently. For example, how are $\tilde{Z}$, $h(\cdot)$, and $q(\cdot)$ defined? What is $m$ in Theorem 1, and which sparsity constraints Theorem 2 refers to? Are $\theta$ one-dimensional factors (parameters)? What is $u$ in Eq. 3?  This is problematic, as, e.g., I failed to understand the proof of Theorem 1. Some statements might be even wrong, given the current version of the paper. For example, I do not understand, how the dimensions of $X$ and $Z$ can be different and $g(\cdot)$ is an invertible transformation, and at the same time, the determinant of Jacobian is well-defined (App. A). In the case of the dimensions mismatch, an invertible transformation would be non-continuous (see Netto's theorem). On the other hand, if we assume a same-dimensional manifold embedded in higher-dimensional space, the change of variables formula used in App. A would be different, e.g., see [3].

**Realisticness of the assumptions**. The major discussion is missing on whether the assumptions in Theorems 1 and 5 are realistic. For example, how could one assume the dimensionality of $Z$ only given $X$? Or even when assuming it, how can we verify, given the data from several environments, whether the number of environments is sufficient? I would love to see a real-world application or a case—study, where we could make those assumptions, or at least speculate about them.

**Experimental evidence/evaluation issues**. The provided experiments aim to support the claims of Theorem 3, and I did not find any regarding Theorem 5 for identifying latent factors. The provided experiments still contain very little implementation and evaluation details, and there is no source code available. For example, it is not clear, how the synthetic data is sampled, e.g., what are “noises” and “biases” in Sec. 5.3? Additionally, I don’t understand, how the provided synthetic datasets satisfy the assumptions of Theorem 3, e.g., what is the sufficient number of environments, what is the dimensionality of $X$, or how we ensure that the MLP is an invertible function.  On the other hand, the evaluation itself is ambiguous, as it is unclear, how the scatter plots in Figures 2 and 3 support the claim that “the hidden structure is recovered”.

Given all the mentioned issues, I tend to reject the paper. I looking forward to the rebuttal and don’t mind increasing my score if the issues are resolved.

References:
[1] Immer, Alexander, et al. "On the identifiability and estimation of causal location-scale noise models." International Conference on Machine Learning. PMLR, 2023.
[2] Zhang, Kun, and Aapo Hyvarinen. "On the identifiability of the post-nonlinear causal model." arXiv preprint arXiv:1205.2599 (2012).
[3] Rezende, Danilo Jimenez, et al. "Normalizing flows on tori and spheres." International Conference on Machine Learning. PMLR, 2020.

**Questions:**

- Traditional SCMs, defined by Pearl [1], assume that latent variables by definition do not have parents. Why cannot we directly model an SCM for $X$ without assuming another latent SCM over $Z$ and a nonlinear mixing function? I would like to hear a discussion on this, as this modelling approach seems to be way simpler.
- Why is $Z_i$ a function of the $\hat{Z}_k$ or $\hat{Z}_l$, if $Z$ and $\hat{Z}$ are observed and modelled variables, respectively, and, thus, are variables of different SCMs? What is $\frac{dZ_i}{d \hat{Z}_k}$ in App. A?

References:
[1] Bareinboim, Elias, et al. "On Pearl’s hierarchy and the foundations of causal inference." Probabilistic and causal inference: the works of Judea Pearl. 2022. 507-556.

---

> ### Author Response · Authors · 2023-11-21
> **Detailed Responses to Your Comments (1/2)**
>
> We sincerely thank the reviewer for the time devoted and the thoughtful comments. Please find the response to your comments and questions below.
>
> Q1: Connections to previous works.
>
> A1: Thanks for the question. We fully agree with you that the connection between independence and second-order derivatives of the log-density was the key to the identifiability of some functional causal models. At the same time, the task we are trying to address is different from causal discovery, since we need to first identify the latent causal variables and then discover the hidden structure, all from some general nonlinear mixtures of these causal variables, i.e., causal representation learning. To the best of our knowledge, our proof technique has never been introduced in this literature.
> Furthermore, as you correctly mentioned, we are dealing with a setting that might be more general than existing identifiable SCMs, since we do not have similar constraints on the functional relations and we allow both the causal variables and structures to not be observed. At the same time, these two settings are different since we are incorporating additional information such as sufficient changes in the distribution.
>
> Q2: Clarification of theorems and notations.
>
> A2: Thanks for your suggestions and sorry for the potential confusion and typos. We have incorporated the clarification into the updated manuscript. Specifically, we have made the following clarifications/corrections:
>
> - (1) $\tilde{Z}$ should be $\hat{Z}$, which denotes the estimated latent variables.
> - (2) $h(\cdot)$ denotes the mapping between the estimated latent variables $\hat{Z}$ and the true variables $Z$. This is also the case for $q(\cdot)$. For consistency, we have changed them to $h$ or $h^{-1}$.
> - (3) $m$ should be $u$, which denotes the auxiliary variable.
> - (4) $\theta$ represents the latent factor that govern the change of hidden causal mechanisms. $u$ in Eq. 3 denotes the auxiliary variable.
> - (5) For Theorem 1, we do not need to assume that $g$ is invertible as long as the function between estimated latent variables $\hat{Z}$ and the true variables $Z$ (i.e., $h$) is invertible, since the change of variable formula only applies between $p(\hat{Z};\theta)$ and $p(Z;\theta)$. Similar to previous work (e.g., [1,2]), $g$ only needs to be injective. We have corrected it in the manuscript.
>
> Apologies again for the potential confusion. We sincerely appreciate your detailed suggestions, which are very helpful in improving the clarity of our manuscript.
>
> Q3: Realisticness of the assumption.
>
> A3: Thanks for your question. As mentioned in A2(5), we do not assume that $Z$ and $X$ are of the same dimension, similar to many previous works in causal representation learning (e.g., [1, 2]). And the determination of the dimension in practice is a typical model selection problem. In practice, one may extend model selection method such as cross-validation or sparsity regularization on a mask to determine the number of dimension of $Z$ [9]. The requirement of a sufficient number of environments is also standard in the literature (see e.g. a recent survey [3]). Various real-world experimental results (e.g., visual disentanglement [5], domain adaptation [6], video analysis [7], and image-to-image translation [8]) in this line of research indicate that it is likely to hold in practice. Meanwhile, for many tasks, the number of environments is actually accessible, just like the number of domains in transfer learning and the number of time indices in time series data. We have updated the manuscript to hightlight the related discussion.

---

> ### Author Response · Authors · 2023-11-21
> **Detailed Responses to Your Comments (2/2)**
>
> Q4: Experiments and evaluations.
>
> A4: Thanks for pointing this out. We assume that the influences from the domain and its parents are location and scaling transformations. Specificically, for each component $z_i$ in each domain, suppose the parents are $PA(i)$, the $z_i=\sum_{j\in PA(i)} s_{j,i}z_j+b_i$, where $s_{j,i}$ are scalars sampled from $Unif[0.5, 2]$ and $b_i\in Unif[-2,2]$. We use 100 domains such that the number of environments is sufficient. As for the nonlinear mixing function, we follow the standard nonlinear ICA literature and construct an MLP whose output dimension is the same as the input dimension. Each layer of MLP consists of an orthogonal matrix weight and a leaky-relu function. Since the weights are orthogonal and the activation is invertible, the MLP is invertible too. For the Y-structure, we have $z_1\rightarrow z_3 \leftarrow z_2$ and $z_3 \rightarrow z_4$. As shown in the Fig. 3(a)(b), the estimated $\hat{z}_1$ recovers $z_2$ and $\hat{z}_1$ is independent from $z_1$; the estimated $\hat{z}_2$ recovers $z_1$ and $\hat{z}_2$ is independent from $z_2$. Therefore, our estimated $\hat{z}_1, \hat{z}_2$ recovers the $z_2, z_1$ and their independence relationship. As for $\hat{z}_3$, it is dependent on $\hat{z}_1$ and $\hat{z}_2$. Therefore, we have $\hat{z}_1\rightarrow \hat{z}_3 \leftarrow \hat{z}_2 \Leftrightarrow z_2 \rightarrow z_3 \leftarrow z_1$. As for $\hat{z}_4$, it is dependent on all previous variables, i.e., $\hat{z}_1, \hat{z}2, \hat{z}_3$. Since we have the sparsity regularization, we can remove the redundant link from $\hat{z}_1$ and $\hat{z}_2$ to $\hat{z}_4$. Therefore, we are able to recover the true latent cause structure. In addition, in eq 6, we have an adjacency matrix to learn, the learned adjacency matrix is identical to the true adjacency matrix, which further supports our theory.
>
> Q5: The necessity of learning latent SCMs and the nonlinear mixing function.
>
> A5: We appreciate the great question, and would like to briefly summarize the discussion as follows:
>
> - (1) Latent causal variables are crucial because they represent unmeasured or unmeasurable influences. For example, in social science research, a latent variable might represent an abstract concept (e.g., peer support in teacher’s burnout study [4]) that influences but is not directly measurable through observed variables.
> - (2) It is common to have dependencies among those abstract concepts in practice, thus allowing an SCM among them (which also covers the empty graph) instead of assuming all variables are independent is much more general and can be applied in more real-world scenarios.
> - (3) In addition, in order to identify the latent SCM, we need to first identify those latent causal variables $Z$ from only observed variables $X$. Since we do not want to put strong constraints on the relations between $Z$ and $X$, we just assume that the function $g$ in $x = g(z)$ is nonlinear, which is the nonlinear mixing function.
>
> We have updated the manuscript in order to better highlight the practical significance of the considered general setting.
>
> Q6: Why is $Z_i$ a function of the $\hat{Z}_k$ and $\hat{Z}_l$? What is $\frac{\partial Z_i}{\partial \hat{Z}_k}$?
>
> A6: Thanks for pointing out this very interesting finding. In order to identify those latent causal variables $Z$, we would like to make sure that our estimated variables $\hat{Z}$ recovered the ground-truth $Z$ up to some mild indeterminacies. Without that guarantee, $\hat{Z}_k$ can be a mixture of an arbitrary set of latent causal variables in $Z$, making the identification less meaningful. This is similar to the task of disentanglement and we provided theoretical guarantees of the disentangled results. $\frac{\partial \hat{Z}_i}{\partial Z_k} \neq 0$ means the estimated variable $\hat{Z}_i$ is not a function of $Z_k$, removing $Z_k$ from the mixture.
>
> References:
>
> [1] Buchholz, Simon, et al. "Learning Linear Causal Representations from Interventions under General Nonlinear Mixing." NeurIPS 2023
>
> [2] Khemakhem, Ilyes, et al. "Variational autoencoders and nonlinear ica: A unifying framework." ATSTATS 2020
>
> [3] Hyvärinen, Aapo, Ilyes Khemakhem, and Hiroshi Morioka. "Nonlinear independent component analysis for principled disentanglement in unsupervised deep learning." Patterns 2023.
>
> [4] Byrne, Barbara M. Structural equation modeling with Mplus: Basic concepts, applications, and programming. routledge, 2013.
>
> [5] Khemakhem, Ilyes, et al. "Ice-beem: Identifiable conditional energy-based deep models based on nonlinear ica." NeurIPS 2020
>
> [6] Kong, Lingjing, et al. "Partial Identifiability for Domain Adaptation." ICML 2022
>
> [7] Yao, Weiran, et al. "Learning temporally causal latent processes from general temporal data." ICLR 2022
>
> [8] Xie, Shaoan, et al. "Unpaired Image-to-Image Translation With Shortest Path Regularization." CVPR 2023.
>
> [9] Xie, Shaoan, et al. "Multi-domain image generation and translation with identifiability guarantees." ICLR 2022.

---

> > ### Comment · Reviewer_mmWQ · 2023-11-23
> >
> > Thank you for your response and clarifications! Yet, I still think there are unresolved issues in the paper.
> >
> > Nevertheless, the uploaded version of the paper seems to be different from what the authors described in their response, and there are some new unexplained changes, too. For example,  $m$ was not actually changed with $u$. Also, the proof of Theorem 1 (now it is Lemma 1) is different from the original formulation, as now it does not have $g(\cdot)$ at all or does not use its injective property.
> >
> > Also, I am still confused between the dimensionality mismatch of  $X$ and $Z$. If function $g(\cdot)$ is injective and continuous, then it effectively sets the dim. of $Z$ higher or equal to dim. of $X$. Or do we allow for non-continuous functions as well?
> >
> > After the revision, I wonder what is $\hat{g}^{-1}$, if we do not assume that neither $g$ nor $\hat{g}$ are invertible functions but only injective.
> >
> > As such, I still tend to keep my score the same.

---

> > > ### Author Response · Authors · 2023-11-23
> > > **Thanks for your prompt response**
> > >
> > > Thank you so much for your prompt response. We are very grateful for your further feedback, which has helped us identify some typos that led to the confusion. We have made the following changes for clarification:
> > >
> > > 1. In the current manuscript, $m$ has been changed to $u$.
> > >
> > > 2. Regarding the confusion about the dimensionality of $X$ and $Z$, we realized it was actually caused by a typo. The dimension of $Z$ should be $n$ and that of $X$ should be $d$. So the dimension of $Z$ is actually smaller or equal to that of $X$. We are very sorry about this and have corrected it in the current manuscript. Specifically, the updated part of the manuscript is as follows:
> > >
> > > > Let $X=(X_1,\dots,X_d)$ be a $d$-dimensional random vector that represents the observations (e.g., images). As is standard in literature [1,2, 3], we assume that they are generated by $n$ hidden causal variables $Z=(Z_1,\dots,Z_n)$ via a nonlinear injective mixing function $g:\mathbb{R}^n\rightarrow\mathbb{R}^d$ ($d \geq n$), which is also a $\mathcal{C}^2$ diffeomorphism.
> > >
> > > Note that this is a standard assumption in related literature, see e.g. Sec. 3.1 in [1], Sec  4.2 in [2], and Assumption 1 in [3]. Since our proof is based on the compound of mixing and demixing functions, we do not have to introduce $g$ in the proof, which is also the case in previous work.
> > >
> > > We apologize again for the potential confusion caused by these typos. We hope our clarification and further updates resolve the remaining issues.
> > >
> > > ---
> > > References:
> > >
> > > [1] Khemakhem, Ilyes, et al. "Variational autoencoders and nonlinear ICA: A unifying framework." ATSTATS 2020
> > >
> > > [2] Hälvä, Hermanni, et al. "Disentangling identifiable features from noisy data with structured nonlinear ICA." NeurIPS 2021
> > >
> > > [3] Buchholz, Simon, et al. "Learning linear causal representations from interventions under general nonlinear mixing." NeurIPS 2023

---

> > > > ### Comment · Reviewer_mmWQ · 2023-11-23
> > > >
> > > > Thanks for another  clarification, but now you claim that $g(\cdot)$ is a diffeomorphism but not just an injection? If this is the case, it is also an invertible map, which sets the dimensions of X and Z equal. If not, I’m confused: what is $\hat{g}^{-1}$ in your paper?

---

> ### Author Response · Authors · 2023-11-23
> **Thanks for your feedback; the dimensions of X and Z are not necessarily the same**
>
> Dear Review mmWQ,
>
> Many thanks for the feedback!  In our setting, X might have higher dimensions and redundant information (for instance, consider image pixels).  Consider the following example in the simple linear case.  Let
> $$
> \mathbf{X} = \begin{bmatrix} 1 & 0 \newline 1 & 1 \newline  1 & 1 \end{bmatrix} \mathbf{Z}, $$
> where the transformation from $\mathbf{Z}$ to $\mathbf{X}$ is 3-by-2, but there still exists a mapping from $\mathbf{X}$ to $\mathbf{Z}$.
>
> What do you think?
>
> Thanks once again,
>
> Authors of #444

---

### Meta-Review · Area_Chair_XpJf · 2023-12-06

**Metareview:**

The paper tackles observational causal discovery, and focuses on identifying "the hidden causal variables, their causal relations, and the latent factors governing the changes in the causal mechanisms" based on multiple distributions.
Key ingredients in the theoretical analysis are: the statement that deriving twice the probability distribution w.r.t. 2 independent variables is 0 (Lin 97); the fact that the derivative of the log likelihood w.r.t. the different parameters $\theta_i$ accounting for the diverse distributions form independent vectors.
Some reviewers appreciate the generality of the setting and the results; for most reviewers however, a detailed discussion was required to understand the contributions and limitations. The experimental validation is not effective in showing the impact of the contributions.
The area chair encourages the authors to polish and publish this very promising paper.

**Justification For Why Not Higher Score:**

-

**Justification For Why Not Lower Score:**

N/A

---

### Decision · Program_Chairs · 2024-01-16

Reject